# SNX9-induced membrane tubulation regulates CD28 cluster stability and signalling

**Manuela Ecker[1], Richard Schregle[2,3], Natasha Kapoor-Kaushik[4], Pascal Rossatti[2], Verena M Betzler[2], Daryan Kempe[1], Maté Biro[1], Nicholas Ariotti[4,5], Gregory MI Redpath[1]\*, Jeremie Rossy[1,2,3]\***

[1]EMBL Australia Node in Single Molecule Science, School of Medical Sciences and the ARC Centre of Excellence in Advanced Molecular Imaging, University of New South Wales, Sydney, Australia; [2]Biotechnology Institute Thurgau (BITg) at the University of Konstanz, Kreuzlingen, Switzerland; [3]Department of Biology, University of Konstanz, Konstanz, Germany; [4]Electron Microscopy Unit, Mark Wainwright Analytical Centre, University of New South Wales, Sydney, Australia; [5]Institute for Molecular Bioscience (IMB), University of Queensland, Brisbane, Australia

**Abstract** T cell activation requires engagement of a cognate antigen by the T cell receptor (TCR) and the co-stimulatory signal of CD28. Both TCR and CD28 aggregate into clusters at the plasma membrane of activated T cells. While the role of TCR clustering in T cell activation has been extensively investigated, little is known about how CD28 clustering contributes to CD28 signalling. Here, we report that upon CD28 triggering, the BAR-domain protein sorting nexin 9 (SNX9) is recruited to CD28 clusters at the immunological synapse. Using three-dimensional correlative light and electron microscopy, we show that SNX9 generates membrane tubulation out of CD28 clusters. Our data further reveal that CD28 clusters are in fact dynamic structures and that SNX9 regulates their stability as well as CD28 phosphorylation and the resulting production of the cytokine IL-2. In summary, our work suggests a model in which SNX9-mediated tubulation generates a membrane environment that promotes CD28 triggering and downstream signalling events.

**\*For correspondence:**
gregmi.redpath@gmail.com (GMIR);
jeremie.rossy@bitg.ch (JR)

**Competing interest:** The authors declare that no competing interests exist.

## Editor's evaluation

Efficient T cell activation requires co-engagement of the T cell receptor (TCR) and of the CD28 co-stimulatory molecules. Both the TCR and CD28 form micro clusters at the plasma membrane of activated T cells. The present study is important in that it showed that by inducing membrane tubulation, the BAR domain-containing protein SNX9 generates a membrane environment that promotes CD28 triggering.

## Introduction

T cell activation is at the centre of the adaptive immune response and relies primarily on the T cell receptor (TCR). Priming of naive T cells requires an additional costimulatory signal delivered by CD28 (*Boomer and Green, 2010*; *Esensten et al., 2016*; *Rudd et al., 2009*). Upon binding of the ligands CD80 and CD86 (otherwise known as B7-1 and B7-2) at the surface of professional antigen-presenting cells, CD28 recruits several effector proteins including filamin-A, Grb2, PI-3K, and PKCθ. Ultimately, CD28 triggering promotes T cell activation through cytoskeletal rearrangements, the nuclear translocation of the transcription factor NFAT and the secretion of internleukin-2 (IL-2) (*Sanchez-Lockhart*

*et al., 2004*; *Fraser et al., 1991*). In absence of CD28 signalling, naive T cells fail to activate and fall into a permanent state of unresponsiveness known as anergy. Anergy is characterised by the inability to produce IL-2 and proliferate (*Harding et al., 1992*; *Jenkins et al., 1988*). CD28 signalling also promotes cytokines transcription independently of TCR (*Raab et al., 2001*).

T cell activation by antigen-presenting cells leads to the formation of a specialised cell-cell interface, the immunological synapse (*Rossy et al., 2012*; *Davis and Dustin, 2004*). Stimulation of TCR and CD28 is required to localise CD28 within the immunological synapse (*Sanchez-Lockhart et al., 2008*), whose formation is impaired in absence of CD28 co-simulation (*Wülfing et al., 2002*; *Huang et al., 2002*). Furthermore, localisation and activation of CD28 at the synapse are required to trigger downstream signalling events required for productive T cell activation (*Sanchez-Lockhart et al., 2008*; *Tseng et al., 2005*; *Yokosuka et al., 2008*; *Huang et al., 2002*).

Within the immunological synapse, surface receptors, signalling and adaptor proteins assemble in protein clusters, which are essential for the initiation and amplification of T cell signalling (*Campi et al., 2005*; *Varma et al., 2006*; *Rossy et al., 2012*). Accordingly, T cell activation or engagement of CD80/86 leads to CD28 clustering, which is driven only by interactions within the extracellular domain of CD28 (*Tseng et al., 2005*; *Tseng et al., 2008*; *Yokosuka et al., 2008*). Importantly, TCR and CD28 clusters are spatially segregated within the immunological synapse (*Tseng et al., 2005*; *Tseng et al., 2008*), which suggests that they might have distinct properties and regulatory mechanisms. How CD28 cluster stability is regulated and how these clusters contribute to CD28 signalling and T cell activation remains poorly understood.

Sorting nexin 9 (SNX9) is a BAR-domain protein that stands at the interface between the plasma membrane and intracellular membranes (*Bendris and Schmid, 2017*). SNX9 binds to specific areas of membranes through two types of domains: a curvature-sensitive BAR domain (*Shin et al., 2008*) and a phosphoinositides-interacting pleckstrin homology (PX) domain (*Pylypenko et al., 2007*). The affinity of SNX9 for distinct membrane domains is coupled with strong membrane remodelling abilities, inherent to the membrane-bending activity of the BAR domains and because of interactions with membrane remodelling proteins. The SH3 domain of SNX9 recruits N-WASP, an activator of the Arp2/3 actin nucleating complex (*Yarar et al., 2007*; *Shin et al., 2008*) and the membrane scission protein dynamin (*Lundmark and Carlsson, 2003*; *Shin et al., 2008*). The so-called low complexity domain of SNX9 mediates further interactions with components of the actin polymerisation and clathrin-mediated endocytosis machineries such as clathrin, AP-2 and Arp2/3 (*Lundmark and Carlsson, 2003*; *Shin et al., 2008*). Unidentified domains of SNX9 also link it to the small GTPases RhoA and Cdc42 (*Bendris et al., 2016*). Hence, SNX9 activity is related to localised actin polymerisation and formation of tubular invaginations of the plasma membrane (*van Weering et al., 2012*; *Yarar et al., 2007*; *Shin et al., 2008*), often in the context of clathrin-mediated endocytosis (*Lo et al., 2017*; *Schöneberg et al., 2017*). SNX9 plays a role in many cellular processes that involve membrane remodelling, such as the formation of the apical domain of polarised cells (*Román-Fernández et al., 2018*), macropinosome formation (*Wang et al., 2010*), phagosome maturation (*Almendinger et al., 2011*), or invadopodia formation (*Bendris et al., 2016*).

SNX9 expression is decreased in T cells upon inflammation (*Ish-Shalom et al., 2016*) and has been implicated in CD28 endocytosis (*Badour et al., 2007*). In this study, we sought to investigate how the membrane remodelling properties of SNX9 contribute to organise the immunological synapse, T cell activation and CD28 signalling. We show that T cell activation leads to the formation of tubules decorated by SNX9 at the immunological synapse. Our investigations further indicate that these tubules are connected to CD28 clusters at the plasma membrane and that SNX9 regulates the stability of CD28 clusters. Finally, we present functional data showing that SNX9 is required for CD28 phosphorylation, subsequent NFAT nuclear translocation and IL-2 production upon T cell activation. Together our data indicate that membrane organisation at CD28 clusters by SNX9 promotes CD28 triggering and contributes to steer T cells away from anergy during activation.

## Results

### Recruitment of SNX9 to the immunological synapse upon T cell activation

SNX9 has been shown to support membrane remodelling during various cellular processes (*Bendris and Schmid, 2017*; *Lundmark and Carlsson, 2009*) and is expressed in T cells (*Ish-Shalom et al., 2016*). However, its role in T cell activation remains to be demonstrated. We first investigated if SNX9 is recruited to the immunological synapse upon T cell activation. To do so, Jurkat T cells were activated with Raji B cells pulsed with or without superantigen staphylococcal enterotoxin E (±SEE respectively) and fixed at 2, 5, 10, or 15 min after initial cell-cell contact. Conjugates were stained with an antibody against SNX9 and phalloidin-Alexa488 to detect F-actin to identify the synapse (*Hammer et al., 2019*). A blinded analysis of the localisation of SNX9 revealed that it was present at the immunological synapse from 2 min onwards, where it was detected in 75% ± 7.6% of the conjugates after 15 min of activation (*Figure 1A and B*).

This result was confirmed with Jurkat T cells expressing SNX9-mCherry and TCR$\zeta$-EGFP activated on cover glass coated with antibodies against CD3ε and CD28 and imaged live from 15 min after activation. 3D reconstructions showed that SNX9 was enriched close to the cover glass at the immunological synapse, identified by the signal of TCR$\zeta$-EGFP (*Figure 1C*). We further confirmed the recruitment of SNX9 to the immunological synapse in primary T cells interacting with dendritic cells (*Figure 1—figure supplement 1*). CD4+ T cells were isolated from transgenic OT-II mice, virally transduced to expressed SNX9-GFP and activated with LPS-matured bone-marrow-derived dendritic cells (BMDCs) pulsed with the OT-II peptide. We observed a clear recruitment to the synapse, here identified by CD28, in over 70% of the CD4+ T cells. One notable difference compared to the data obtained in Jurkat cells is the fact that a bit more than 10% of the unspecific conjugates displayed enrichment of SNX9 where CD4+ T cells contacted the BMDCs (no SNX9 was detected at the Jurkat-Raji cell interface in absence of SEE, *Figure 1B*). Together, these results confirm the recruitment of SNX9 to the immunological synapse upon T cell activation.

*Figure 1C* further revealed that the signal of SNX9-mCherry was organised in membranous structures immediately above the synapse. To better understand how these SNX9-positive structures relate to T cell activation, we tested how TCR and/or CD28 signalling promote their formation. We imaged z-stacks of live Jurkat T cells expressing SNX9-GFP and deposited on non-activating (PLL-coated) or on activating (coated with antibodies against CD3ε and/or CD28) cover glass (*Figure 1D*). The number of SNX9-positive structures in the 3D reconstructions of the cells was quantified using a particle detection analysis routine based on intensity thresholding (*Figure 1—figure supplement 2*; *Figure 1E*). Strikingly, SNX9 was completely cytosolic in resting cells, where not a single SNX9-positive structure was detected over four independent experiments. This transition from cytosolic to membrane at the immunological synapse upon activation was also observed in primary CD4+ T cells activated by BMDCs (*Figure 1—figure supplement 1*). Of note, the low levels of activation reported for PLL in T cells (*Santos et al., 2018*) were not sufficient to promote membrane association of SNX9. Stimulation of CD28 alone and co-stimulation of CD28 and TCR were about twice as potent as TCR triggering alone in stimulating the formation of SNX9-positive structures. Altogether, these data show that CD28 stimulation during T cell activation promotes the formation of SNX9-positive structures at the immunological synapse.

Activation leads to the localisation of the T cell intracellular trafficking machinery behind the immunological synapse (*Onnis and Baldari, 2019*). To determine if the SNX9-positive membranous structures resulting from CD28 triggering are linked to synaptic endosomes, we characterised their spatial distribution in respect to vesicles and endosomes positive for TCR or for the early endosome marker Rab5. To do so, the whole volume of activated Jurkat T cells expressing SNX9-GFP or mCherry and either TCR$\zeta$-mCherry, Rab5-mCherry or CD28-GFP was imaged live with an axial resolution of 190 nm using an Airyscan detector (*Figure 1F*). The number of structures positive for mCherry was determined in each z-stack to obtain their axial distribution relative to the first focal plane where the cell initially came in to focus (defined as axial position = 0 μm; *Figure 1G*). In line with our previous observations (*Compeer et al., 2018*; *Redpath et al., 2019*), we detected many TCR$\zeta$ and Rab5-positive vesicles and endosomes distributed through the cell, from slightly above the synapse to the top of the cell, with their number peaking at 1.35 and 1.8 μm from the 0 μm position, respectively. The signal of CD28 was restricted to the plasma membrane of the immunological synapse (peak at

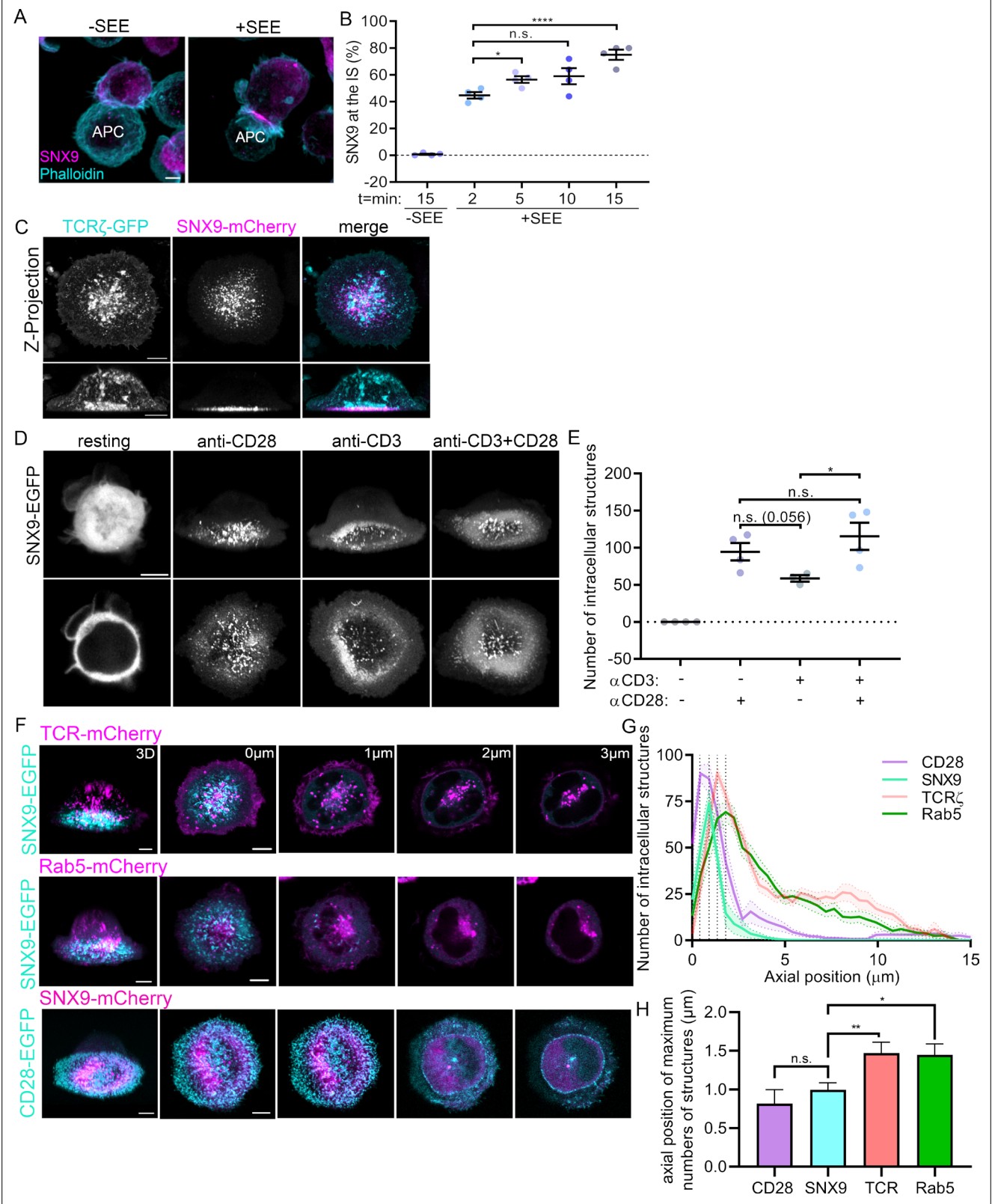

**Figure 1.** Recruitment of SNX9 to the immunological synapse upon T cell activation. (**A**) Representative images of maximum intensity projections from Z-stacks (0.19 µm per slice) of WT Jurkat T cells conjugated to SEE pulsed (+ SEE) or non-pulsed (-SEE) antigen presenting cells (APC). Cells were then fixed after incubation at 37 °C for 2, 5, 10, or 15 min, permeabilized and stained for endogenous SNX9 and actin (Phalloidin-488). (**B**) Mean percentage of SNX9 recruitment towards the synapse at indicated time points of cell shown in A identified in a blinded analysis of 60–100 cells per condition in four

*Figure 1 continued on next page*

*Figure 1 continued*

independent experiments. (**C**) Representative live images of WT Jurkat T cells activated on cover glass (anti-CD3ε and anti-CD28), expressing SNX9-mCherry and TCR ζ -EGFP. Top panel: maximum intensity projections from Z-stacks (0.19 µm per slice) Bottom panel: shows 3D reconstructions. (**D**) Representative images of 3D reconstruction and maximum intensity projections of wildtype Jurkat T cells expressing SNX9-EGFP either in resting (PLL-coated cover glass) or under different activating conditions. (**E**) Mean number of SNX9-positive structures in T cells activated under different conditions in three to four experiments (8–11 cells per experiment). (**F**) Representative live images of activated Jurkat T cells co-expressing SNX9-EGFP or -mCherry and either TCR ζ -mCherry (top), Rab5-mCherry (middle) or CD28-EGFP (bottom). (**G**) Quantification of the number of SNX9-EGFP/-mCherry, TCR ζ -mCherry, Rab5-mCherry and CD28-EGFP positive vesicles in each frame of the z-stack of three to four experiments (5 cells per experiment). (**H**) Average axial position where the maximum number of CD28, SNX9, TCR ζ and Rab5 structures were identified. Dots represent individual experiments. Error bars indicate mean ± SEM. n.s. = not significant, * = p < 0.05, **** = p < 0.0001 from Student's T-test of means of independent experiments. Scale bar = 5 µm.

The online version of this article includes the following figure supplement(s) for figure 1:

**Figure supplement 1.** SNX9 is recruited to the immunological synapse of primary T cells where it co-localises with CD28.

**Figure supplement 2.** Worflow for identification of intracellular structures.

0.4 µm from the 0 µm position). This suggests that the CD28-positive structures identified by the analysis routine were not vesicles but rather plasma membrane microclusters. Strikingly, the membranous structures positive for SNX9 were detected just above the immunological synapse within a narrow peak around 0.9 µm from the 0 µm position, immediately after the peak defined by the CD28 clusters and yet closer to the synapse than TCR or Rab5 endosomes. We further extracted the axial position of the maximum number of structures detected for each cell of each condition (*Figure 1H*). This analysis confirmed that SNX9 intracellular structures were below TCR and Rab5-positive endosomes. While the average of the positions of the maximum number of structures for SNX9 was higher than for CD28, the difference was not statistically significant. However, the lack of significance is more likely to reflect the limitation of the resolution of the system rather than an absence of difference, as detailed below in Figure 4. In summary, these experiments show that SNX9 is not associated with endosomes in resting cells but rather decorates structures strictly proximal to the immunological synapse upon T cell activation.

## Recruitment of SNX9 to CD28 clusters at the immunological synapse

As our observations showed that CD28 triggering leads to the formation of SNX9-positive membrane structures at the immunological synapse to a higher extent than TCR stimulation alone, we sought to determine the potential connection between SNX9 and CD28 clusters. Jurkat T cells expressing wild type CD28 (CD28WT) or non-phosphorylatable CD28 generated by point mutations of intracellular tyrosine residues to phenylalanine (CD28YF) fused to EGFP were activated on cover glass coated with antibodies against anti-CD3ε and CD28 antibodies, fixed and stained with an antibody against SNX9. We imaged these cells under total internal reflection fluorescence (TIRF) illumination for optimal visualisation of protein distribution within the plasma membrane (*Figure 2A*). In accordance with previous publications (*Tseng et al., 2005*; *Tseng et al., 2008*; *Yokosuka et al., 2008*), both CD28WT and CD28YF were organised in clusters within the plasma membrane. Crucially, SNX9 was also clustered with line profiles of fluorescence intensity across the cell (*Figure 2B*). We used an intensity-based threshold analysis identifying clusters in each channel to confirm that the majority of clusters defined by SNX9 overlapped with CD28 clusters (*Figure 2C*; *Figure 2—figure supplement 1*). While not significant, we observed a clear tendency of a lower overlap of SNX9 clusters with the non-phosphorylatable form of CD28 when compared to WTCD28, suggesting that CD28 phosphorylation may contribute to the recruitment of SNX9. Of note, the same intensity-based threshold revealed no obvious overlap between SNX9 and Rab5, which we used here as negative control for the quantification approach. Photoactivation of CD28-PAmCherry at the plasma membrane by TIRF 405 nm illumination also revealed that CD28 clusters remain co-localised with clusters of SNX9-EGFP after photoactivation (*Figure 2—figure supplement 2*).

Lateral mobility and membrane deformation, which can influence receptor localization, are impaired when T cells are activated by immobilised antibodies. To circumvent this limitation, we used Airyscan confocal microscopy and 3D reconstruction to visualise protein clustering in immunological synapses of conjugates between Jurkat T cells and Raji B cells pulsed with SEE (*Figure 2—figure supplement 3*). En face projections revealed that these synapses contained large clusters of co-localising CD28

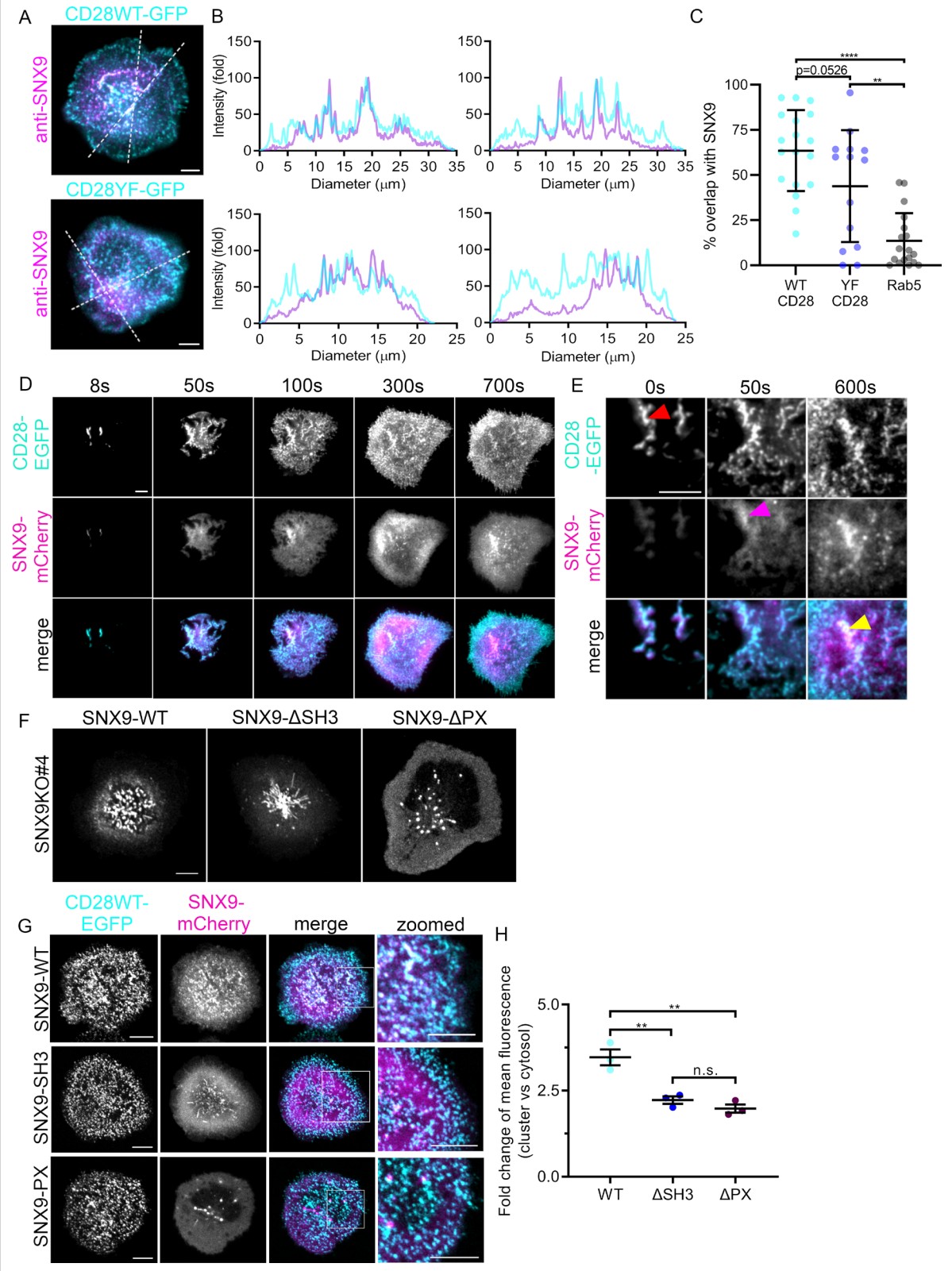

**Figure 2.** Recruitment of SNX9 to CD28 clusters upon T cell activation. (**A**) Representative TIRF images of fixed WT Jurkat T cells expressing CD28WT-EGFP or CD28YF-EGFP (cyan) and activated for 10 min. (**B**) Fold fluorescent intensities of CD28WT/YF-EGFP (cyan) and anti-SNX9 (purple) along the doted lines in A. (**C**) Quantification of the percentage overlap between WT or YF CD28 and SNX9, or Rab5 and SNX9. (**D**) Representative TIRF live images of WT Jurkat T cells expressing SNX9-mCherry and CD28-EGFP at indicated timepoints. Images representative of four independent experiments

*Figure 2 continued on next page*

*Figure 2 continued*

including four to eight cells each (**E**) Representative zoomed images of C, showing CD28-EGFP (red arrow) arriving at the immunological synapse first, recruiting SNX9-mCherry (magenta arrow) over time to the same microclusters (yellow). (**F**) Representative live images of activated Jurkat SNX9 KO#4T cells expressing SNX9-WT-mCherry (left), SNX9-ΔSH3-mCherry (middle) or SNX9-ΔPX-mCherry (right). (**G**) Representative live images of activated SNX9 KO#4T cells co-expressing CD28WT-EGFP and SNX9WT-mCherry, SNX9-ΔSH3-mCherry or SNX9-ΔPX-mCherry. (**H**) Fold change of mean fluorescence intensity profiles of SNX9-WT-mCherry, SNX9-ΔSH3-mCherry or SNX9-ΔPX-mCherry intensity within CD28WT-EGFP microclusters compared to the mean fluorescence intensity profile of SNX9WT-mCherry within the cytosol. Error bars indicate mean ± SEM. n.s. = not significant, ** = p < 0.01, from Student's T-test of means of three independent experiments of 6–10 cells, or individual cells (**C**). Scale bar = 5 µm.

The online version of this article includes the following video and figure supplement(s) for figure 2:

**Figure supplement 1.** Representative confocal images from three independent experiments of WT Jurkat T cells expressing SNX9-GFP (cyan) and mCherry-Rab5 (magenta) and activated for 10 min.

**Figure supplement 2.** SNX9 is recruited into CD28 microclusters upon T cell activation.

**Figure supplement 3.** SNX9 is recruited into CD28 microclusters upon T cell activation by antigen-presenting cells.

**Figure supplement 4.** SNX9 expression in Jurkat WT and SNX9KO T cells.

**Figure 2—video 1.** Representative live cell movie showing SNX9 recruitment to CD28 clusters during T cell activation.

https://elifesciences.org/articles/67550/figures#fig2video1

and SNX9, confirming the results obtained using antibody-coated glass and TIRF microscopy. Finally, colocalization of SNX9 and CD28 at the synapse was further confirmed in primary CD4⁺ T cells activated by BMDCs (*Figure 1—figure supplement 1*). Together, these data indicate that SNX9 at the immunological synapse is contained in CD28 clusters in a dynamic fashion.

We next investigated the temporal dynamics of the formation of CD28 and SNX9 clusters to understand which protein recruits the other. To do so, T cells expressing CD28WT-EGFP and SNX9-mCherry were imaged in TIRF while being deposited on cover glass coated with activating antibodies against CD3ε and CD28 (*Figure 2D*; *Figure 2—video 1*). Cells were imaged from the moment they touched the glass until they were fully activated (12 min in total). CD28 clusters were first observed in 'ridges' of plasma membrane mediating the first contacts of the cell with the stimulatory surface, as previously described for clusters of Lat in activating T cells (*Sherman et al., 2011*). SNX9 was first found in the cytosol within these ridges and clusters of SNX9 were detected approximately 50 s later (*Figure 2E*). This observation is consistent with the transition of SNX9 from cytosol to membrane that we observed upon T cell activation (*Figure 1D*). In line with the results obtained with endogenous SNX9 in fully spread and fixed cells (*Figure 2A and B*), the SNX9 signal first appeared within the denser CD28 clusters and remained within these clusters for the entire duration of the acquisition (*Figure 2—video 1*). Thus, these time-sequence experiments establish that SNX9 is recruited to pre-existing CD28 clusters, as opposed to CD28 being recruited to SNX9 clusters. These results further indicate that SNX9 is not required to establish CD28 clusters upon T cell activation.

Because SNX9 was mobilised to CD28 clusters independently of the phosphorylation of CD28 intracellular domains, we set out to investigate the interactions responsible for this recruitment. SNX9 contains a membrane binding domain that binds to phosphoinositide-rich (PX) and curved (BAR) membranes (*Pylypenko et al., 2007*) and an SH3 domain that mediates interactions with proteins containing proline-rich motifs, including dynamin and the Arp2/3 activator WASP (*Yarar et al., 2007*; *Shin et al., 2008*). In order to determine if these interaction domains were responsible for the recruitment of SNX9 into CD28 clusters, we generated two deletion mutants preventing SNX9 interactions with most of its protein partners – missing the SH3 domain (ΔSH3) – or with membranes – missing the PX domain (ΔPX) (*Pylypenko et al., 2007*; *Schöneberg et al., 2017*). We further generated SNX9 knock-out (SNX9 KO) Jurkat cell lines using CRISPR/Cas9 gene editing (*Figure 2—figure supplement 4*). When expressed in SNX9 KO cells, SNX9-ΔSH3-mCherry defined long tubules, while SNX9-ΔPX-mCherry was cytosolic and in highly mobile vesicles (*Figure 2F*). Crucially, none of these mutants displayed a clustered distribution. These observations are in line with previous studies showing that preventing interaction with the actin-polymerising machinery or with membranes results in abnormal tubular structures (*Shin et al., 2008*) and a cytosolic localisation (*Pylypenko et al., 2007*), respectively.

We next used these mutants to determine which functional domains of SNX9 are required for recruitment to CD28 clusters. SNX9 KO Jurkat T cells expressing SNX9-WT, SNX9-ΔSH3 or SNX9-ΔPX fused to mCherry and CD28-EGFP were activated on antibody-coated cover glass (*Figure 2G*). To

quantify the recruitment of SNX9 mutants into CD28-positive clusters, the CD28 signal was used to generate a 'cluster' mask and the fluorescence intensity of the mCherry signal was measured both inside and outside this mask (*Figure 2H*). We measured significantly less mCherry signal within CD28 clusters for both deletion mutants when compared to WT SNX9. Thus, SNX9 recruitment to CD28 clusters is not mediated by phosphorylated tyrosine residues on CD28 but requires the functional domains of SNX9 conferring interaction with proteins containing proline-rich motifs and membranes. As CD28 triggering seems to play a preponderant role in the recruitment of SNX9 to CD28 clusters (*Figure 1D and E*), we can only speculate that this recruitment relies on interactions with CD28 effectors and/or a specific lipid environment within the clusters. Importantly, because the low complexity domain which binds to clathrin and AP-2 (*Lundmark and Carlsson, 2009*) is still present in the ΔSH3-SNX9 mutant, our results suggest that interactions with clathrin or AP-2 are not sufficient to locate SNX9 in CD28 clusters. The fact that the ΔSH3-SNX9 mutant was not recruited to CD28 clusters despite having been shown to be normally recruited to clathrin coated pits (*Schöneberg et al., 2017*) further indicates that SNX9 localisation in CD28 clusters is independent of the clathrin endocytic machinery.

## Dynamics and persistence of SNX9 within CD28 clusters

Our data clearly show that SNX9 is recruited to existing CD28 clusters upon T cell activation. We next used photoactivation of fluorescent proteins to obtain a better insight into the dynamics of how SNX9 associates with CD28 clusters. We designed an approach to distinguish between unconstrained diffusion of the PAmCherry signal from the region of photoactivation throughout the entire synapse (*Figure 3A* upper) and the propagation of this signal only through clusters (*Figure 3A* lower). Activated cells expressing CD28-EGFP and SNX9 fused to photoactivatable mCherry (SNX9-PAmCherry) were repetitively illuminated every 34 s with 405 nm light at the cell periphery in order to evaluate the diffusion of SNX9 through the synapse (*Figure 3A and B*; *Figure 3—video 1*; *Figure 3—figure supplement 1*). The SNX9-PAmCherry fluorescence rapidly spread from the photoactivated region over the whole cell-glass interface. As in *Figure 2G*, we used a mask defined by CD28-EGFP to quantify the distribution of the SNX9-PAmCherry signal within CD28 clusters (*Figure 3C*). This quantification revealed that SNX9 diffused rapidly throughout the immune synapse, while predominantly co-localising with the clustered signal of CD28. The signal of SNX9-PAmCherry was significantly higher in clusters than in the cytosol within 4 s after the beginning of 405 nm illumination. Of note, we did not observe formation of any endosomal structures positive for SNX9-PAmCherry following photoactivation when a threshold and subsequent size-based quantification was used (size cut-off: 5–1000 pixel units, 65 nm per pixel; threshold images and quantification in *Figure 3—figure supplement 2*) as we did when we used the same approach to investigate endocytosis of TCR ζ, flotillins or clathrin (*Compeer et al., 2018*; *Redpath et al., 2019*). This observation suggests that SNX9 at the plasma membrane in CD28 clusters is not incorporated into endocytic vesicles.

We then used a similar photoactivation approach in TIRF to interrogate how stable the association of SNX9 is with CD28 clusters (*Figure 3D*). The plasma membrane of Jurkat T cells expressing CD28-EGFP and SNX9-PAmCherry and activated on antibody-coated cover glass was exposed to 405 nm light under TIRF illumination for 5 s. Images where then taken only at 1 sec, 5 min and 10 min after photoactivation to minimise bleaching of PAmCherry and to allow quantification of the persistence of the PAmCherry signal in CD28 clusters (*Figure 3E and F*). The fluorescence intensity of photoactivated SNX9-PAmCherry in the mask defined by CD28-EGFP did not decrese significantly over a time period of 10 min, indicating that SNX9 is stably associated with CD28 clusters over time. Collectively, these data reveal that SNX9 molecules are rapidly exchanged between CD28 clusters but nevertheless remain strictly incorporated within the population of CD28 clusters.

## SNX9 induces tubulation in CD28 clusters

We have established that SNX9 defines membranous structures proximal to the immune synapse and is incorporated into CD28 clusters in an activation dependent manner. To better understand the interactions and interdependencies of SNX9 and CD28 at the immunological synapse, we analysed how they are spatially organised by three-dimensional correlative light and electron microscopy. First, we performed whole cell imaging of fixed Jurkat T cells expressing CD28-EGFP and SNX9-mCherry with Airyscan confocal microscopy (*Figure 4—video 1*). These dishes were then processed, flat embedded, and thick sections (250 nm) were cut and imaged for electron tomographic analyses.

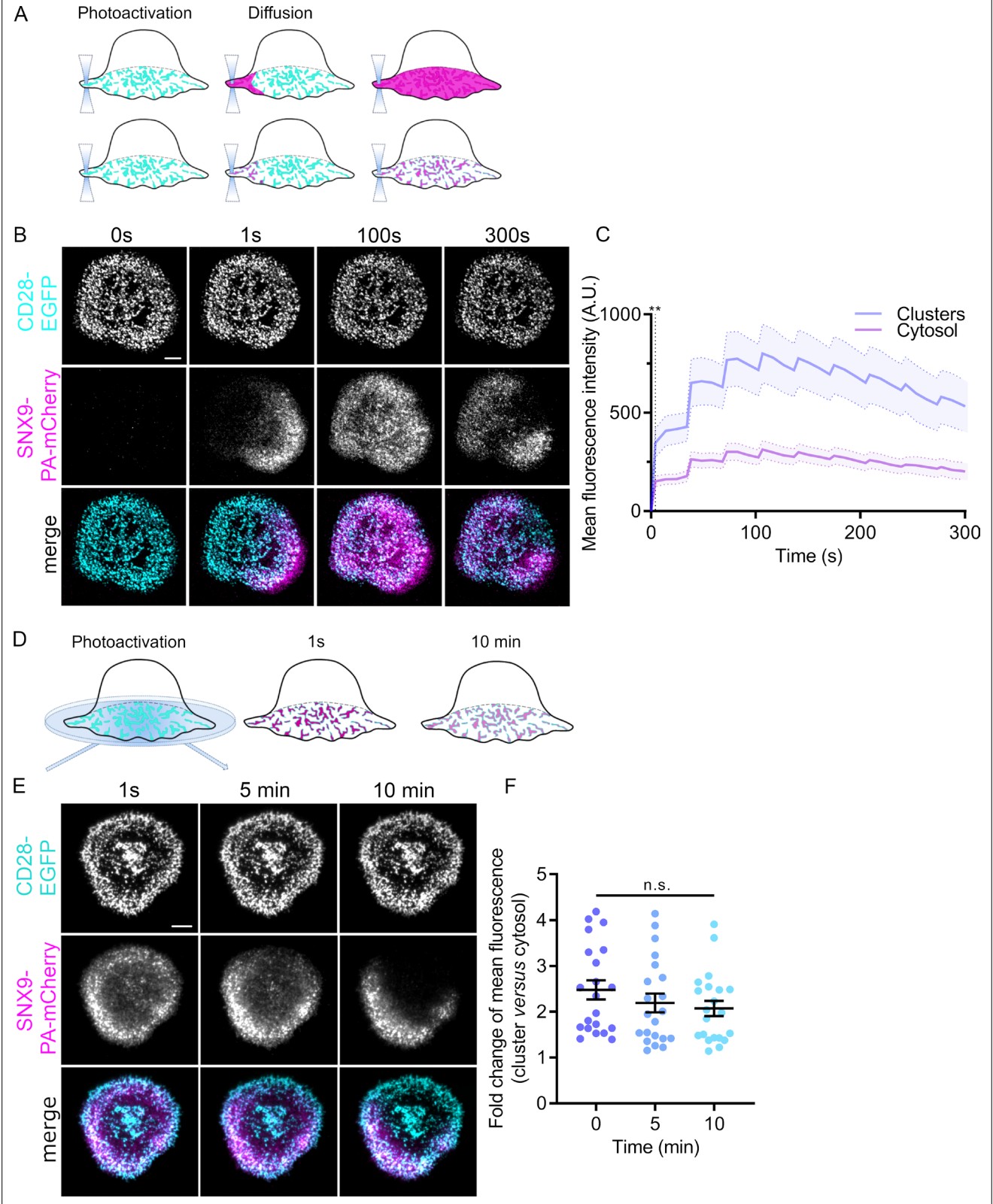

**Figure 3.** Dynamics and persistence of SNX9 within CD28 clusters. (**A**) Schematic of photoactivation of fluorescent proteins at the plasma membrane of cover glass activated cells with the photoactivated protein diffusing either though the cell membrane or through protein-clusters localised at the plasma membrane. (**B**) Representative confocal time series of SNX9-PAmCherry repetitively photoactivated by 405 nm laser at the membrane region of interest (dashed line) in activated SNX9 KO#4 cells. (**C**) Mean fluorescence intensity profiles of SNX9-PAmCherry intensity within CD28-EGFP clusters (blue) or

*Figure 3 continued on next page*

*Figure 3 continued*

within the cytosol (magenta). Dashed line represents the timepoint from which the SNX9-PAmCherry intensity significantly diverges between cytosol and clusters (** = p < 0.01). Error bars = mean ± SEM = means of three independent experiments of 6–10 cells. (**D**) Schematic of whole cell photoactivation in TIRF. (**E**) Representative TIRF images of WT activated Jurkat T cells co-expressing CD28-EGFP and SNX9-PAmCherry photoactivated once for 6 s. (**F**) Fold change of mean fluorescent intensity profiles of SNX9-PAmCherry intensity within CD28-EGFP microclusters compared to the cytosol of three experiments (6–9 cells per experiment). n.s. = not significant. Scale bar = 5 µm.

The online version of this article includes the following video and figure supplement(s) for figure 3:

**Figure supplement 1.** Example of custom-made FIJI quantification of cluster versus cytosol.

**Figure supplement 2.** Quantification approach used to count the number of PAmCherrry-positive vesicles after photoactivation.

**Figure 3—video 1.** Representative movie showing the dynamic localisation of SNX9 in CD28 clusters.

https://elifesciences.org/articles/67550/figures#fig3video1

The cell and region of interest were re-registered in the thick sections (*Figure 4A–D*) and the 3D fluorescence z-stacks were fitted to the 2D electron micrograph of the thick sections (*Figure 4C and D*). Tilt series were acquired, and the 3D reconstruction showed abundant tubules present within the section volume (E i – E iv; *Figure 4—video 2*). 3D correlation showed the CD28 clusters were highly abundant and present at the immune synapse and matched well with the undulations of the plasma membrane (z = 0 nm, *Figure 4F*).

The fluorescence signal of the SNX9 decorated and defined long tubular structures emerging from the immunological synapse within the tomographic volume (z = 100 nm, *Figure 4G*). Strikingly, the SNX9 tubules arose from the CD28 clusters at the plasma membrane (*Figure 4H–J*). These structures had an average diameter between 80 and 100 nm, which is consistent with previous observation made with reconstituted tubules in vitro (*van Weering et al., 2012*). We used IMOD to generate contours to identify and determine the length, surface area and volume of tubules connected to CD28-rich area of the plasma membrane, positive for SNX9 and displaying consistent densities (*Figure 4—figure supplement 1*). SNX9-positive tubules showed a clear tendency to be larger than SNX9-negative tubules, being significantly longer (*Figure 4—figure supplement 1C*). Of note, the average length of SNX9-positive tubules (250 nm) made it impossible to formally distinguish them from the plasma membrane in conventional fluorescence microscopy in *Figure 1H*. In summary, these high-resolution analyses and quantification suggest that SNX9 is enriched in tubules emanating from CD28 clusters on the plasma membrane and promote their elongation in activated T cells. As we did not detect CD28 within these tubular SNX9 structures, we speculate that their function is to regulate the presence of lipids or specific effectors within CD28 clusters, and thereby mediate CD28 signalling as detailed below in the discussion.

## SNX9 promotes CD28 residence in clusters

We then sought to uncover the functional role on T cell signalling of connection of the SNX9-positive tubules and CD28 clusters. To determine if SNX9 contributed to CD28 cluster formation, we quantified the number of clusters in activated Jurkat WT and SNX9 KO cells expressing CD28WT or CD28YF. Mutations of tyrosine residues in the YF mutant had no impact on the number of CD28 clusters as previously published (*Yokosuka et al., 2008*), nor did the knock-out of SNX9 (*Figure 5A*), indicating that SNX9 is not required for CD28 to assemble into clusters. This result is consistent with data presented in *Figure 2D*, which show that SNX9 is recruited to pre-existing CD28 clusters.

As SNX9 is exchanged rapidly within CD28 clusters (*Figure 3A*) we queried if CD28 was exchanged within these clusters by using a similar photoactivation approach. Surprisingly, we observed that when photoactivated with 405 nm light at the periphery of the cells, CD28-PAmCherry diffused rapidly through the entire activation area while remaining strictly within clusters (*Figure 5C*; *Figure 5—video 1*). By contrast, TCR $\zeta$-PAmCherry underwent moderate diffusion within the plasma membrane before being quickly internalised within endocytic vesicles as previously reported (*Figure 5C*; *Figure 5—video 1*; *Compeer et al., 2018*). This result reveals that even though CD28 clusters appear very stable (*Figures 2 and 3*), CD28 is constantly exchanged within the population of clusters.

Considering that CD28 clusters are not static structures, we investigated whether SNX9 might have an influence on cluster stability using fluorescence recovery after photobleaching (FRAP). Regions of interest corresponding to CD28-EGFP clusters were photobleached by illumination with an 800-nm

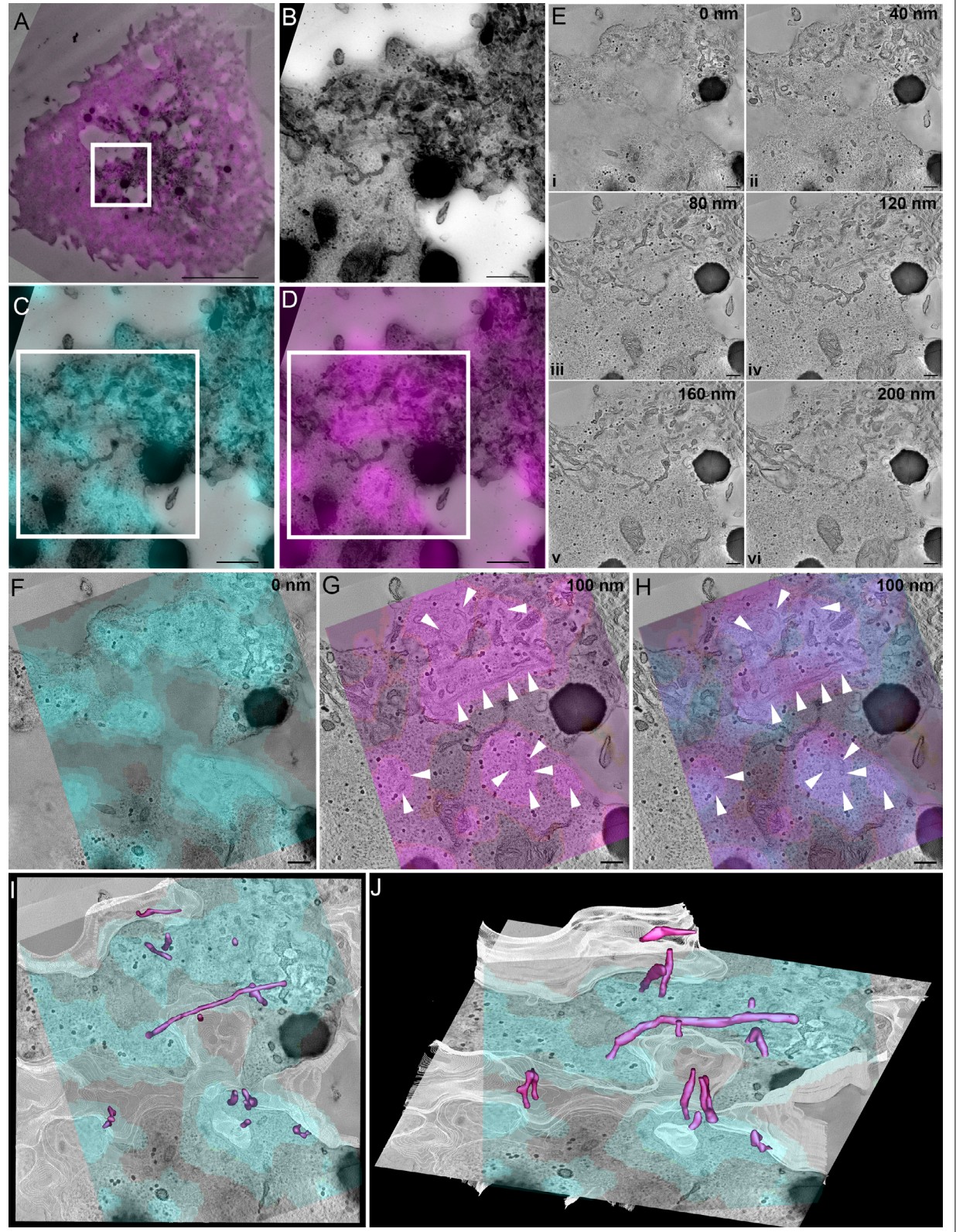

**Figure 4.** SNX9 defines tubules that are connected to CD28 clusters. (**A**) Correlation of SNX9-mCherry transfected cell with the same cell re-registered by transmission electron microscopy. White box: region of interest positive for CD28-GFP and SNX9-mCherry. Scale 5 µm. (**A**) Region of interest from (**A**) identified from high magnification stitched transmission electron micrographs at ×10,000 magnification. White box = area of tomographic analysis. Scale 500 nm. (**C**) Overlay of CD28-EGFP with stitched electron micrograph. Scale 500 nm. (**D**) Overlay of SNX9-mCherry with stitched electron micrograph.

*Figure 4 continued on next page*

*Figure 4 continued*

Scale 500 nm. (**E**) Optical slices from a reconstructed tomogram of different z-depths demonstrating abundant tubules emerging from the plasma membrane. Scale 200 nm. (**F**) Overlay of CD28-EGFP signal at the cell-glass interface on the very base of the cell. Scale 200 nm. (**G**) Overlay of SNX9-mCherry signal with an electron tomographic slice deeper into the cell shows abundant tubulation (white arrowheads) at SNX9-mCherry-positive areas. Scale 200 nm. (**H**) Overlay of CD28 and SNX9 signal suggesting SNX9-tubules (white arrowheads) emerge from CD28-positive clusters. Scale 200 nm. (**I, J**) Rotated views of a segmented volume showing SNX9-positive tubules (magenta) directly connect with the immunological synapse/plasma membrane (white) at the CD28-positive clusters (cyan).

The online version of this article includes the following video and figure supplement(s) for figure 4:

**Figure supplement 1.** Identification and quantification of tubular structure in correlative electron tomography.

**Figure 4—video 1.** Confocal slices of the WT Jurkat cell expressing CD28-EGFP and SNX9-mCherry used to overlaid on the EM pictures.

https://elifesciences.org/articles/67550/figures#fig4video1

**Figure 4—video 2.** Optical slices from a reconstructed tomogram of different z-depths demonstrating abundant tubules emerging from the plasma membrane.

https://elifesciences.org/articles/67550/figures#fig4video2

two-photon laser in WT or SNX9 KO Jurkat T cells expressing CD28WT-EGFP or CD28YF-EGFP with or without SNX9 overexpression (*Figure 5D*, *Figure 5—video 2*). We used two-photon illumination to ensure that only molecules within the focal plane of the plasma membrane were bleached. The increase of fluorescence over time in the bleached clusters was normalised by the signal from unbleached ROIs to correct for fluorescent protein bleaching. The final curves of fluorescence recovery were obtained by calculating the mean of all the individual acquisitions (*Figure 5E*).

Divergence analysis revealed that the recovery rate of fluorescence in CD28 clusters of cells over-expressing SNX9 was significantly lower than the recovery rate of control cells from 94 s on. Plotting the mean of the fluorescence intensity profiles at 380 s after initial photobleaching further showed that CD28WT-EGFP clusters had recovered 57.4% ± 10.9% of their initial fluorescence in WT cells without SNX9 transfection and only 34.6% ± 4.8% of fluorescence in cells overexpressing SNX9 (*Figure 5F*). This suggests that SNX9 promotes the stability of CD28 clusters. In line with this, fluorescence intensity within CD28 clusters recovered significantly faster in both SNX9 KO Jurkat T cell lines, diverging significantly from the WT control recovery curve at 124 and 384 s after bleaching in two SNX9 KO clones. The recovery curve of KO#4 diverged from the WT curve only at the latest time points, but it nevertheless showed impaired recovery relative to WT cells. To verify the divergence of the KO#4 from the WT, we normalised the curves and fitted them with a bi-exponential decay, indicating that the fluorescence recovery occurred via two processes characterised by kinetic constants tau1 and tau2 (*Figure 5—figure supplement 1*). The kinetic constant of the component, tau2, was significantly different from the WT for both KO#3 and KO#4, confirming that the recovery of fluorescence within CD28 clusters was affected by SNX9 knock-out. This was further illustrated by the fact that the percentage of CD28-EGFP fluorescence recovery at 380 s was higher in SNX9 KO cells than for WT cells (57.4%), reaching 70.4% ± 4.7% and 69% ± 13%. Of note, these differences were not the result of differential expression of CD28, as the absence of SNX9 in the knock-out cell lines did not alter CD28 expression (*Figure 6*). There was no significant difference in the FRAP of clusters of CD28WT and CD28YF, indicating that unlike SNX9, phosphorylation of the intracellular domains of CD28 does not contribute to CD28 cluster stability.

Finally, we measured free diffusion in the plasma membrane using the myristoylated and palmitoylated membrane anchor of the kinase Lck, Lck10-EGFP. In contrast to the fluorescence recovery for CD28 (57.4%), the fluorescence signal recovered in bleached areas within seconds and reached 89.2% ± 4.4% of the maximal signal at 380 s (*Figure 5—figure supplement 2*). More importantly, there were no significant differences in the FRAP of Lck10 upon overexpression or knock-out of SNX9, confirming the specificity of the effect of SNX9 on CD28 cluster stability. Altogether, these data strongly suggest that SNX9 contributes to regulate the rates of exchange of CD28 in and out of clusters and thereby their stability.

## Surface levels and endocytosis of CD28 upon SNX9 knock-out

SNX9 has been shown to play a role in clathrin-mediated endocytosis by promoting the recruitment of dynamin and possibly actin-nucleating machinery to the constricted neck of budding endocytic vesicles (*Bendris and Schmid, 2017*; *Lundmark and Carlsson, 2009*). Notably, a study reported SNX9

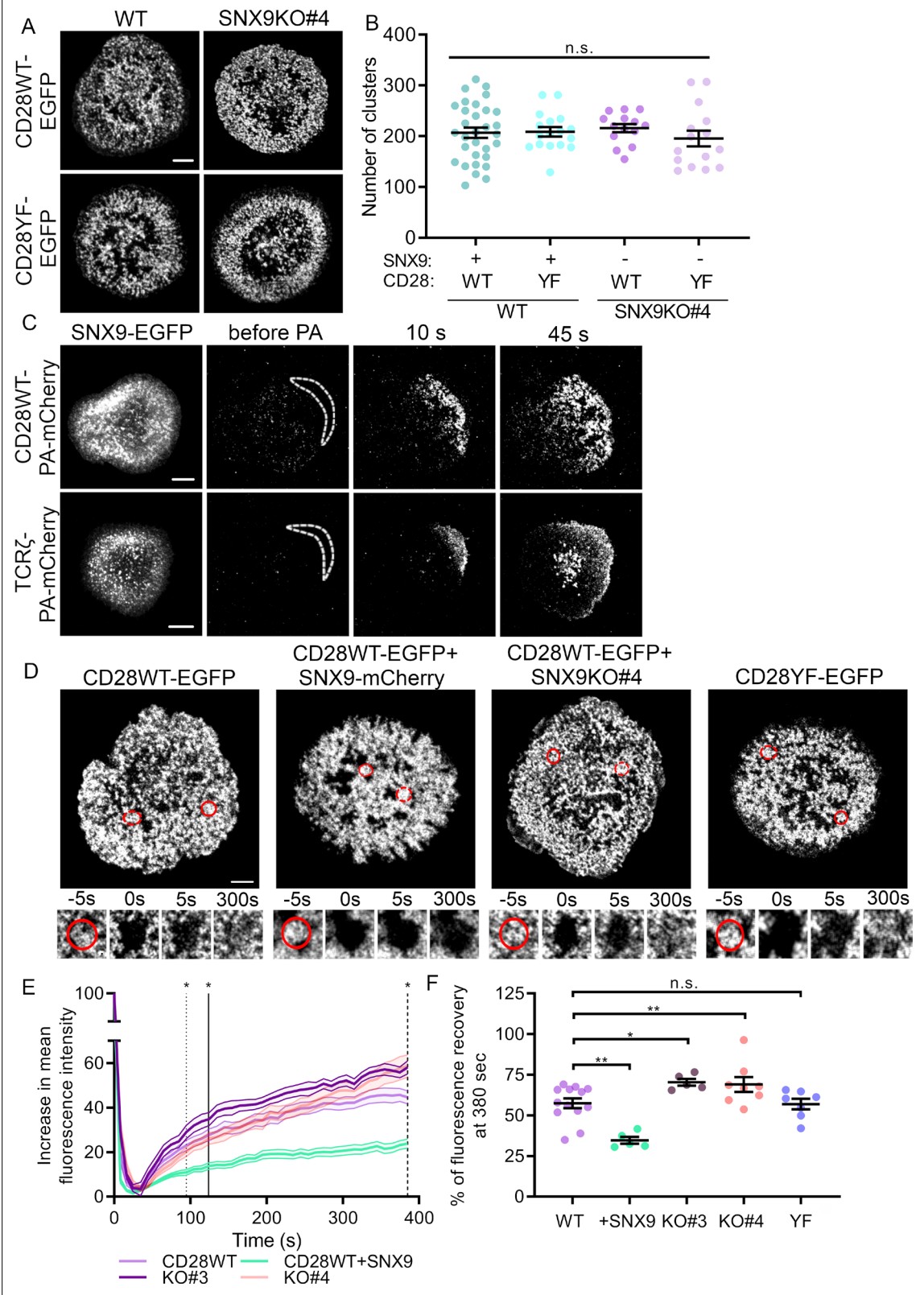

**Figure 5.** SNX9 promotes CD28 cluster stability. (**A**) Representative live images of activated WT Jurkat (left) and SNX9 KO#4 (right) T cells expressing CD28WT-EGFP or CD28YF-EGFP with or without SNX9WT-mCherry. (**B**) Number of CD28WT/YF-EGFP positive structures in Jurkat SNX9 KO T cells with or without SNX9WT-mCherry expression. Dots represent individual cells (15-28) of three to five independent experiments (**C**) Representative images at indicated time points of activated WT Jurkat T cells expressing SNX9-EGFP and CD28WT-PAmCherry (top panel) or TCR ζ -PAmCherry (bottom

*Figure 5 continued on next page*

*Figure 5 continued*

panel) photoactivated within the dashed regions. Images representative of three and four independent experiments including three to seven cells each. (**D**) Representative images of activated WT and SNX9 KO#4 Jurkat T cells expressing CD28WT or YF-EGFP with or without SNX9-mCherry and photobleached in the region of interest (red circle). Red discontinuous circles represent non-beached control regions. (**E**) Increase in mean fluorescence intensity over time of bleached regions shown in (**D**). (**F**) Mean fluorescence intensity at t = 380 s from (**E**). Lines = time point of significant divergence (p < 0.05) from CD28WT (dotted:+ SNX9; solid: KO#3; dashed: KO#4). Data from 5 to 13 independent experiments involving three to six cells each. Dots on chart indicate independent experiments. Error bars indicate mean ± SEM. n.s. = not significant, * = p < 0.05, ** = p < 0.01 from Student's T-test of mean of independent experiments. Scale bar = 5 μm.

The online version of this article includes the following video and figure supplement(s) for figure 5:

**Figure supplement 1.** Fitting of the Fluorescence recovery after photobleaching (FRAP) curves for CD28 in WT, SNX9KO and SNX9 overexpressing Jurkat T cells.

**Figure supplement 2.** SNX9 has no impact on the recovery of Lck fluorescence intensity after FRAP.

**Figure 5—video 1.** Representative movie showing the diffusion of CD28 through clusters (identified by SNX9-mCherry).
https://elifesciences.org/articles/67550/figures#fig5video1

**Figure 5—video 2.** Representative movie showing two-photon FRAP of CD28-EGFP clusters.
https://elifesciences.org/articles/67550/figures#fig5video2

---

and CD28 to be in the same endocytic vesicles and the overexpression of SNX9 promoted CD28 internalisation (*Badour et al., 2007*). Thus, we measured CD28 surface expression and uptake using flow cytometry in order to test if SNX9 could be related to any CD28 endocytic process.

Jurkat WT and SNX9 KO cells were activated with anti-CD3ε and anti-CD28 for 20 min or left in non-activating conditions. Following activation, surface bound antibodies were removed by acid washing and cells were stained with FITC-labelled anti-CD28 to detect surface CD28 (*Figure 6A* and *Figure 6—figure supplement 1*). To investigate CD28 uptake, cells were incubated with anti-CD3ε and anti-CD28-FITC on ice to allow antibody binding, and then at 37°C for 20 min to allow for internalisation. Surface bound antibodies were then washed off by an acid wash and fixed (*Figure 6B*). Cell surface expression and internalisation of CD28 did not differ significantly between Jurkat WT and SNX9 KO in resting or activated cells. To compensate for the difference in CD28 surface expression between the various KO cell lines, we calculated the normalised ratio between surface expression and uptake of CD28-FITC fluorescence signal, which confirmed that SNX9 does not regulate CD28 internalisation from the cell surface in Jurkat T cells (*Figure 6C*).

Our data globally suggest that extensive endocytosis of CD28 does not occur in activated T cells. We did not observe any obvious intracellular pool of CD28 or intracellular vesicles or compartments positive for CD28 in activated cells expressing CD28-EGFP (*Figure 1F*), CD28-PAmCherry was not detected in endocytic vesicles following photoactivation at the plasma membrane (*Figure 5C*; *Figure 3—figure supplement 2*). Furthermore, the low number of intracellular structures positive for CD28 in activated cells matches the lower rates of internalisation of CD28 upon activation when compared to TCR. After 10 min of activation only 42.8% ± 5.7% of the initial levels of cell surface TCR could still be detected at the cell surface, in contract to 88.8% ± 7.3% for CD28 (*Figure 6—figure supplement 2*). In summary, our observations indicate that SNX9 does not significantly contribute to the moderate internalisation of CD28.

We next investigated if SNX9 contributes specifically to CD28 signalling to determine the functional consequences of the regulation of CD28 cluster stability by SNX9. CD28 phosphorylation is the first detectable event in CD28 triggering. It is maximal around 5 min after activation and persists for at least 30 min (*Sadra et al., 1999*). WT and SNX9 KO Jurkat T cells were activated for 5 min in 96-well plates coated with anti-CD3ε and anti-CD28, fixed, permeabilised, and stained with antibodies against phosphorylated CD28-Y218 (pCD28) or phosphorylated TCR ζ -Y142 (pTCR). The fluorescence signal was then measured by flow cytometry (*Figure 7A and C*). Mean fluorescence intensity of the WT cells was normalised to 100% and compared to the mean of each group to evaluate statistical significance using a one sample t-test (*Figure 7B and D*). Both SNX9 KO cell lines showed a significant decrease of CD28 phosphorylation (40% ± 19.2%) compared to WT cells. By contrast, the fluorescence intensity of phosphorylated TCR did not differ between WT and SNX9 KO cells. Thus, SNX9 specifically contributes to the phosphorylation of CD28 upon T cell activation.

CD28 signalling is strongly connected to the nuclear translocation of the transcription factor NFAT and the subsequent secretion of the cytokine IL-2 (*Boomer and Green, 2010*). As CD28

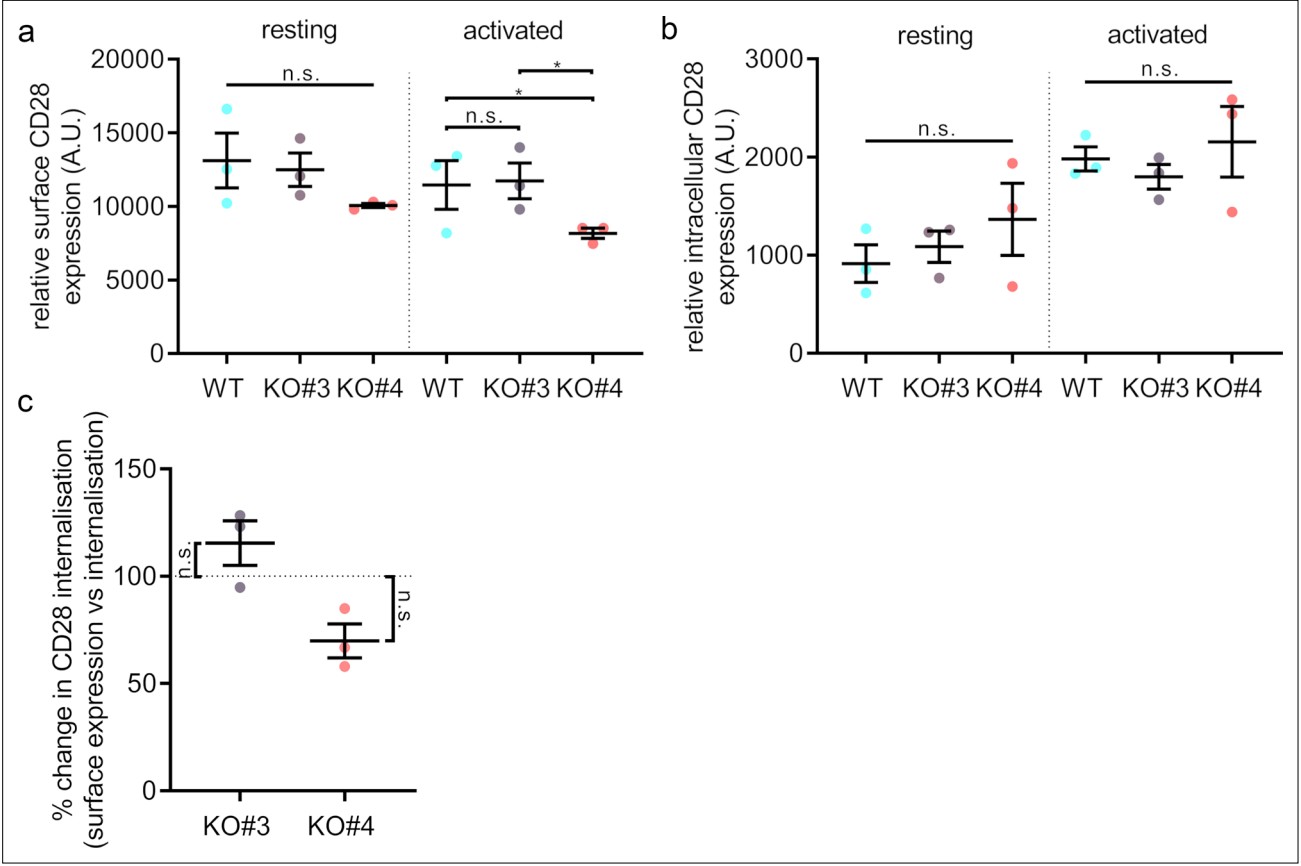

**Figure 6.** Surface levels and endocytosis of CD28 upon SNX9 knock-out. (**A**) Mean fluorescence intensity of resting and activated (20 min) WT Jurkat and SNX9KO#3 and SNX9KO#4T cells incubated with an antibody against CD28 (CD28-FITC). (**B**) Mean fluorescence intensity of resting and activated Jurkat WT and SNX9KO#3 and SNX9KO#4T cells stained with an antibody against CD28 (CD28-FITC) and allowed to internalise CD28 complexes before removing any surface bound antibody. Error bars indicate mean ± SEM. n.s. = not significant, * = p < 0.05 from Student's T-test of means of three independent experiments. (**C**) Ratio of internalised by surface expression of CD28 normalised to WT (100%). Data obtained from three independent experiments, with at least two replicates per experiment. Error bars indicate mean ± SEM. n.s. = not significant, * = p < 0.05, from a one sample T-test comparing values to the normalised WT mean of 100%. SNX9 specifically contributes to CD28-mediated signalling.

The online version of this article includes the following figure supplement(s) for figure 6:

**Figure supplement 1.** Flow cytometry gating strategy.

**Figure supplement 2.** CD28 receptor internalisation very low compared to TCR.

phosphorylation was reduced in SNX9 KO cells, we sought to determine if this had consequences on nuclear translocation of NFAT. WT and SNX9 KO Jurkat T cells were deposited on non-activating or activating cover glass, fixed, permeabilised, and stained with an antibody against NFAT and a nuclear dye (Hoechst) to quantify the level of NFAT translocation to the nucleus (*Figure 7E*). Quantification of the levels of NFAT co-localizing with the nucleus using a dedicated analysis routine (*Schindelin et al., 2012*) revealed that there was significantly less NFAT in the nucleus of SNX9 KO cells compared to WT cells upon activation (*Figure 7F*).

To evaluate the impact of SNX9 knock-out IL-2 secretion, WT or SNX9 KO Jurkat T cells were conjugated with Raji B cells pulsed with or without SEE and incubated for 16 hr in a 96-well plate. The amount of IL-2 present in the supernatant was then analysed by ELISA (*Figure 7D*). We observed a significant decrease of IL-2 secretion from the SNX9 KO cells (an average of 2.20 ng/ml for WT, 1.34 ng/ml for SNX9 KO#3 and 1.17 ng/ml for SNX9 KO#4) compared to WT cells. Taken together, these data on the functional consequences of SNX9 knock-out show that SNX9 is specifically required for CD28 triggering and downstream cellular events upon T cell activation.

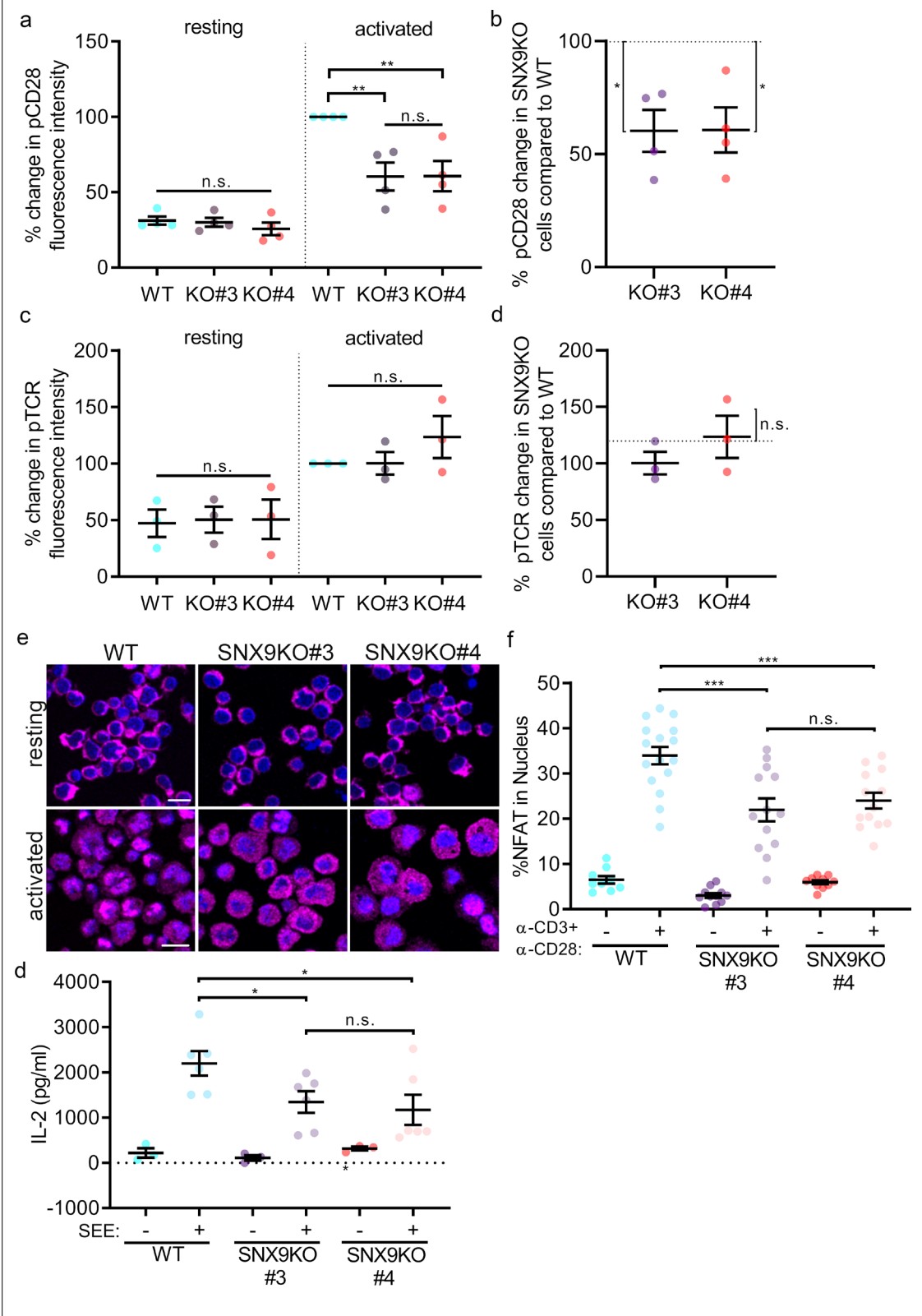

**Figure 7.** SNX9 specifically contributes to CD28-mediated signalling. (**A**) Phosphorylation of CD28 and (**C**) TCR ζ in activated WT Jurkat and SNX9KO T cells detected by phospho-specific antibodies in flow cytometry. (**B**) Normalised (WT = 100%) percentage of pCD28 and (**D**) pTCR ζ in SNX9KO cells compared to WT Jurkat T cells. Dots represent independent experiments (3-4) involving 50'000 cells. Error bars indicate mean ± SEM. n.s. = not significant, * = p < 0.05, from a one sample T-test. (**E**) Representative immunofluorescence images of resting and activated WT and SNX9KO Jurkat T

*Figure 7 continued on next page*

*Figure 7 continued*

cells stained with an antibody against NFAT. (**F**) Integrated NFAT fluorescence intensity over the Hoechst-stained nucleus in WT and SNX9 KO Jurkat T cells activated or not with anti-CD3ε and CD28 antibodies. Dots represent individual fields of view including 22–30 cells each from six independent experiments. Error bars = mean ± SEM. ** = p < 0.001, *** = p < 0.001 from Student's T-test of means of field of views. Scale bar = 20 μm (**G**) WT Jurkat and SNX9KO T-cells were conjugated with SEE pulsed (activated+) or non-pulsed (resting-) Raji B cells and incubated together for 16 hr in a 96-well plate to allow for IL-2 secretion. The supernatant was collected, and the amount of IL-2 present was determined by ELISA. Error bars = mean ± SEM. * = p < 0.05, from Student's T-test of means of 3–6 independent experiments involving $5 \times 10^5$ cells of each type.

## Discussion

T cell activation requires CD28 signalling to complement TCR triggering and induce differentiation and cytokine production (*Boomer and Green, 2010*; *Esensten et al., 2016*; *Rudd et al., 2009*). Requirement for CD28 co-signalling is the reason for the inclusion of the intracellular domain of CD28 in the design of some chimeric antigen receptors in cancer therapies (*van der Stegen et al., 2015*). Like TCR, CD28 assembles into signalling clusters at the immunological synapse (*Tseng et al., 2008*; *Yokosuka et al., 2008*). However, in contrast to TCR, how CD28 clustering contributes to CD28 signalling remains poorly understood. Here, we show that upon CD28 triggering, the membrane-organising protein SNX9 is recruited to CD28 clusters at the immunological synapse where it generates membrane tubulation. Our data further reveal that CD28 is continuously exchanged within the population of clusters, and that SNX9 plays a role in regulating CD28 clusters stability. Functionally, SNX9 brings an essential contribution to CD28 triggering and downstream signalling.

It has been shown that adaptor proteins regulate stability of T cell clusters and thereby determine the intensity of T cell activation (*Lasserre et al., 2011*). The question stemming from our results is how SNX9 achieves a similar function in CD28 signalling. Below we discuss the potential mechanisms through which SNX9 could regulate CD28 triggering and recruitment of effectors to CD28 clusters.

Regulation of cell surface expression modulates receptor signalling and SNX9 plays a role in CD28 internalisation (*Badour et al., 2007*). However, our data suggests very modest rates of CD28 endocytosis occur in activated T cells. Furthermore, SNX9 knock-out had no impact on CD28 surface expression. Another possible mechanism of regulation of CD28 signalling by SNX9 in clusters could be via the control of actin polymerisation, as SNX9 can recruit and activate the actin nucleating complex Arp2/3 through its SH3 domain (*Yarar et al., 2007*). In T cells, actin is required for TCR cluster formation at the immunological synapse (*Campi et al., 2005*) and, more generally, can regulate cluster stability (*Lavi et al., 2012*). CD28 signalling is also connected to actin polymerisation (*Tavano et al., 2006*). Yet, the assembly of CD28 clusters does not involve actin polymerisation (*Yokosuka et al., 2008*) and they have never been described as actin-rich structures.

The SNX9-positive tubular structures that we observed at the immunological synapse rather suggest that SNX9 regulates CD28 signalling by organising the membrane and lipids, perhaps by way of its BAR domain (*Bendris and Schmid, 2017*). Indeed, membrane heterogeneity and local enrichment of lipid species have been shown to regulate many aspects of T cell signalling and activation. Diacylglycerol at the synapse promotes T cell polarity (*Chauveau et al., 2014*) and phosphoinositides control actin organisation (*Le Floc'h et al., 2013*; *Gawden-Bone et al., 2018*) and TCR ζ accumulation (*DeFord-Watts et al., 2011*) at the immunological synapse.

More importantly, phosphatidylserine (*Gagnon et al., 2012*) and phosphoinositides (*Chouaki Benmansour et al., 2018*) play a central role in TCR triggering by favouring the release of intracellular domains of the TCR-CD3 complex from the plasma membrane. This model, in which intracellular domains are inaccessible to effectors while bound to the inner leaflet, has also been put forward to explain CD28 triggering (*Dobbins et al., 2016*; *Yang et al., 2017*). Thus, it is possible that organisation of the plasma membrane by SNX9 within CD28 clusters contributes to establish a distinct lipid environment within CD28 clusters and, by doing so, promotes cluster stability and signalling. Several BAR-domain proteins can cluster phosphoinositides and generate lipid domains (*Picas et al., 2014*; *Zhao et al., 2013*; *Saarikangas et al., 2009*), although this has not yet been described for SNX9. Membrane curvature, like the SNX9-positive tubules that we observed at the synapse, is also thought to promote formation of specific lipid subdomains by driving the enrichment of lipids that preferentially associate with the curvature of the local membrane (*Callan-Jones et al., 2011*; *Parton et al., 2020*). This would favour membrane detachment of the CD28 cytoplasmic domain and, in doing so, would provide improved access of CD28 intracellular domain to effectors.

Clustering of CD28 occurs independently of cytoplasmic domain (*Yokosuka et al., 2008*) and SNX9 was recruited to pre-existing CD28 clusters in our time-lapse experiments. This indicates that SNX9 binding to CD28 is not required for the initial formation of CD28 clusters. However, membrane curvature can favour clustering of proteins (*Reynwar et al., 2007*; *Strahl et al., 2015*). Therefore, membrane curvature induced by SNX9 tubulation in CD28 cluster could promote recruitment and clustering of CD28 effectors to facilitate CD28 phosphorylation and signalling. Aside from clustering, curvature is postulated to influence diffusion of membrane proteins in theoretical models (*Yoshigaki, 2007*; *Balakrishnan, 2000*). We speculate that slower diffusion at CD28 clusters increases the dwell time of CD28 effectors and by doing so potentiates CD28 signalling as it has been recently shown for SOS and N-WASP (*Martin and Mittag, 2019*).

Finally, membrane curvature can directly regulate enzymatic activity, including of proteins that are reported effectors of CD28. The activity of phosphoinositide 3-kinase, which is recruited to the YMNM motif on the cytoplasmic tail of CD28 upon triggering, is increased by membrane curvature (*Bettache et al., 2003*; *Hübner et al., 1998*). Similarly, curvature has been shown to modulate the activity of PKC, another key effector of CD28 signalling (*Goldberg and Zidovetzki, 1998*; *Jiménez-Monreal et al., 1999*; *Epand and Lester, 1990*).

In summary, in this study we show that CD28 clusters at the immunological synapse recruit SNX9 upon CD28 triggering. We present evidence that SNX9 generates membrane tubulation at CD28 clusters and regulates their stability and we establish a clear link between SNX9, and CD28 phosphorylation and downstream signalling in T cells. Our results suggest a revised model of lipid organisation at the immunological synapse, in which membrane-binding protein would organise the composition and curvature of the membrane at T cell microclusters. They further suggest that lipid organisation inside clusters could be central to how signalling is regulated within these structures. This raises many questions. What is the exact nature of the distinct lipid environment promoted by SNX9 within CD28 clusters? Which effectors are either recruited or more likely activated by the CD28-SNX9 subdomains? Answering will require the development of more sensitive probes and approaches to better dissect the lipid environments and their role in regulating signalling in T cell clusters.

## Materials and methods

### Plasmids

Expression constructs encoding for human SNX9-pmCherryC1 (RRID:Addgene_27678), SNX9-EGFP, SNX9ΔPX-mCherry, SNX9ΔSH3-mCherry and CD28-EGFP, were provided by Prof. K. Gaus (University of New South Wales). PAmCherry expression backbone was obtained from Clontech. TCR ζ -Pam-Cherry was made by inserting a PCR product of TCR-PS-CFP2 into pPAmCherry-N1 using EcoRI + AgeI. EGFP-Rab5 was obtained from Addgene (RRID:Addgene_31733).

CRISPR/Cas9 knockouts were generated as described in *Ran et al., 2013*. Briefly, Jurkat T cells were transfected with two guide RNAs designed using the GPP sgRNA Designer (https://portals.broadinstitute.org/gpp/public/analysis-tools/sgrna-design) to target exon 2: AATGAACTGACGGT-TAATGA and exon 3: GAAACATCAAAGGAGAACGA of the SNX9 genomic DNA sequence, together with Cas9-GFP expression plasmid (RRID:Addgene_79144). Twenty-four hours post transfection, single cells were FACS sorted based on GFP signal and seeded into 96-well plates. Cell clones were screened by using western blotting against SNX9 (#181856, 1:4000, Abcam). Clones lacking SNX9 eventually grew to an appropriate population within about 20 days.

### Antibodies

Anti-SH3*P* × 1 (SNX9) (1:100, Abcam Australia Pty Ltd, Cat# 181856), CD28-FITC (1:100, eBioscience, Cat# 11028942), Alexa Fluor 647 anti-CD247 (pY142) (1:20, BD Phosflow, Cat# 558489), Anti-phospho-CD28 (pTyr[218]) (1:20, Merck Cat# SAB4504133), anti-Rabbit Alexa647 (Thermo Fisher Scientific Cat# A-21245).

### Jurkat ell culture and activation

WT T cells (ATCC Cat# TIB-152, RRID:CVCL_0367; tested for microplasma contamination and authenticated by STR profiling), SNX9KO#3 and SNX9KO#4 Jurkat cell lines were cultured in RPMI 1640 medium (Gibco). Additionally, the media was supplemented with 10% (vol/vol) FBS, 2 mM L-Glutamine

and PenStrep (all from Invitrogen). Cells were transfected with 1 µg DNA per 200,000 cells using the Neon electroporation kit (Invitrogen) 18–20 hr prior to imaging.

Before imaging, cells were incubated for 10 min at 37°C on 18-mm glass-coated surfaces (Marienfeld). These were prepared by incubating with 0.001% (vol/vol) poly-L-lysine (P4957; Sigma) (resting condition) for 30 min at room temperature, then washed with 1 x PBS and incubated with 1 µM anti-CD3ε (16–0037; eBioscience) and anti-CD28 (16–0289; eBioscience) antibodies for 1 hr at 37°C (activating condition). For live cell imaging, cells were imaged from 10 to 40 min after their deposition on the cover-glass.

For transferrin internalization, Jurkat T cells were activated by incubation with 1 µM anti-CD3ε and anti-CD28 for 10 min in solution (RPMI) at 37 °C. The media was then exchanged for fresh media with 25 µg.ml$^{-1}$ transferrin Alexa647 (Jackson ImmunoResearch, USA) and incubated for 15 min at 37°C. Cell were then washed with fresh ice-cold media and acid washed to remove any surface bound antibody. Cells were kept on ice and analysed by flow cytometry.

For immunostaining, activated and or resting cells were fixed with 3.7% (vol/vol) EM-grade paraformaldehyde (C004, ProScitech) for 30 min at 37°C. After fixation, cells were permeabilised with 0.15% triton-X100 (Sigma), blocked in 5% BSA for 1 hr and probed with primary and secondary antibodies sequentially.

For cell conjugation, 1 × 10$^6$ /ml Raji B-cells were incubated with 2 µg/ml highly purified staphylococcal enterotoxins (SEE; Toxin Technology Inc) for 1 hr at 37°C. Cells were then washed twice and pelleted by centrifugation (200 x g, 5 min, 37°C) with an equal amount of Jurkat T cells. Subsequently, the mixture of T-cells and B-cells was incubated for 10 min at 37 °C in a water bath. The pellet was carefully resuspended and 0.5 × 10$^6$ cells were deposited on a poly-L-lysine-coated coverslip and allowed to adhere for 15 min. Hereafter, the cells were fixed with 3.7% EM-grade paraformaldehyde for 15 min at RT. For the conjugates used in 3D reconstruction and en face projection, the same procedure was used with the following differences: mCherry-SNX9 and CD28-EGFP were overexpressed and stained with anti-CD28 (mouse; clone E18; 1 µg/mL; Biolegend, Cat# 122002) and anti-SNX9 (rabbit, clone EPR14399; 0.15 µg/mL (1:500); Abcam; Cat# ab181856) and Phalloidin-647 for actin, and included DAPI, which was in the mounting medium (DAPI Fluoromount-G). The second antibodies were anti-mouse coupled to AlexaFluor488 (Invitrogen, Cat# A-11017) and anti-rabbit coupled to AlexaFluor568 (Invitrogen, Cat# A 21069).

## Isolation and transduction of mouse T cells

Transgenic CD4+ T cells expressing the TCR specific for the ISQAVHAAHAEINEAGR (AA 323 339) peptide of ovalbumin in the context of H2-I-Ab MHCII molecules were isolated from OT-II mice (Genotype: B6.Cg-Tg(TcraTcrb)425Cbn/J) using the MojoSort Mouse CD4 T Cell Isolation Kit (Biolegend, Cat# 480033). Isolated T cells were seeded at 1 × 10$^6$ cells/mL in complete RPMI supplemented with 40 U/mL murine IL-7 (Peprotech, Cat# 217–17) and 100 U/mL murine IL 12 (Peprotech, Cat# 210–12). The T cells were activated using Dynabeads Mouse T-Activator CD3/ CD28 for T-Cell Expansion and Activation (Thermo Fisher Scientific, Cat# 11,452D) at a ratio of 1 bead per 2 cells. After 24 hr of activation, cells were transduced with ecotropic retroviral vectors encoding GFP-SNX9 at an MOI of 50–100 in presence of 50 µg/mL protamine sulphate by spinfection (2000 g, 90 min, 32 °C). The T cells were incubated further at 32 °C in the centrifuge for 30 min after centrifugation. Then we added 1 mL fresh complete RPMI supplemented with 40 U/mL murine IL-7 and 100 U/mL murine IL 12 and transferred the cells back into the CO2 incubator. After another 24 hr, the T cells were transduced a second time under the same conditions. Finally, we added 3 mL of complete RPMI supplemented with 40 U/mL murine IL-7 and 50 U/mL human IL 2 (Peprotech, Cat# 200–02), and the cells incubated further at 37 °C. On day 5 of culture, dead cells and the activation beads were removed by density gradient centrifugation using Ficoll-Paque Plus (Cytiva, Cat# GE17-1440-03) The cells were cultured further at 1 × 106 cells/mL in fresh complete RPMI supplemented with 40 U/mL murine IL-7 and 50 U/mL human IL 2. On day 6 of culture, GFP+ T cells were enriched by sorting on a BD FACSAria llu cell sorter (BD Biosciences). Sorted T cells were rested overnight in fresh complete RPMI in presence of 40 U/mL murine IL-7 and 50 U/mL human IL 2.

## Generation of bone-marrow-derived dendritic cells

Bone marrow-derived dendritic cells (BMDCs) were generated from wild type C57BL/6 J mice as described before (*Lutz et al., 1999*). Briefly, we rinsed the bone marrow from femurs and tibiae and washed the cells with PBS. Contaminating erythrocytes were lysed in ACK buffer (8.29 g/L NH4Cl, 1 g/L KHCO3, 0.1 mM EDTA) for 5 min at room temperature (RT) and subsequently washed with PBS supplemented with 5 mM EDTA and 2% iFCS (heat-inactivated FCS). Finally, we seeded the bone marrow cells at $4 \times 10^5$ cells/mL in bacteriological 10 cm petri dishes in 10 mL of RPMI 1640 supplemented with 10% iFCS, 50 µM β mercaptoethanol, 100 U/mL penicillin, and 100 µg/mL streptomycin (complete RPMI 1640) and 200 U/mL murine granulocyte/macrophage-colony stimulating factor (GM CSF; Peprotech, Cat# 315–03). On day 3 of culture, 10 mL of fresh complete RPMI 1640 supplemented with 200 U/mL murine GM-CSF were added. On days 6 of culture, 8.5 mL of the culture medium was removed, and 10 mL of fresh complete RPMI 1640 supplemented with 200 U/mL murine GM-CSF were added. Fully differentiated immature BMDCs were obtained on day 8. To induce maturation of BMDCs, we harvested floating and loosely adherent cells, resuspended the cells in 10 15 mL fresh complete RPMI supplemented with 200 U/mL murine GM-CSF, and added 0.1 µg/mL lipopolysaccharide (LPS; Merck, Cat# L4391) for 20 24 h.

## Conjugation of primary T cells and BMDCs

For the formation of conjugates, BMDCs were incubated with 1 µM of OT-II peptide (Merck, Cat# O1641) or not for 1 hr at 37 °C. The BMDCs were then washed twice with warm complete RPMI and cell numbers adjusted to $1 \times 10^6$ cells/mL. The OT-II T cells were washed with warm complete RPMI and adjusted to $1 \times 10^6$ cells/mL. Both cell types were placed into the CO2 incubator in a 15 mL falcon tube for 10 min to warm up again to 37 °C before conjugation. In a 2 mL reaction tube $1 \times 10^6$ OT-II T cells were mixed with $5 \times 10^5$ BMDCs and the volume filled to 2 mL. The cells were pelleted at 1000 g for 30 s and subsequently incubated at 37 °C in a pre-warmed waterbath for 15 min. Then, the cells were gently resuspended and 1 mL of the cells seeded onto 12-mm coverslips placed into the wells of a 12-well plate. The coverslips were coated with 0.01% poly L lysine (Sigma-Aldrich, Cat# P4832) and 5 µg/mL human fibronectin (purified from human plasma in-house) for 30–60 min at 37°C. Subsequently the 12-well plate was centrifuged for 2 min at 90 g to settle the cells. The cells were allowed to fully settle and warm up for 10–13 min at 37°C, followed by processing for confocal imaging.

## Retroviral vector construct and production

The cDNA of SNX9 was inserted into the retroviral backbone pMSCV-EGFP by restriction enzyme cloning. The resulting plasmid pMSCV-EGFP-SNX9 expressed SNX9 fused to the C terminus of EGFP. Retroviral plasmids were amplified using NEB Stable Competent *E. coli* (High Efficiency) (NEB, Cat# C3040H). Plasmid sequences were verified by sequencing (Microsynth AG). We generated retroviral particles by triple transfection of HEK293T cells with the expression plasmid pMSCV-EGFP-SNX9, the envelope plasmid pEco (part of the Retro-X Universal Packaging System; Takara, Cat# 631530), and the packaging plasmid pUMVC (pUMVC was a gift from Bob Weinberg RRID:Addgene_8449) using polyethylenimine (linear PEI; MW 25,000; Polysciences Europe, Cat# 23966) at 1:5 plasmid to PEI ratio. The cell culture supernatant containing the retroviral particles was collected 48 hr and 72 hr after transfection. Prior to purification, remaining plasmid DNA was removed by digestion with Dnase I (Roche, Cat# 10104159001). After sterile-filtration (0.45 µm, PES membrane), retroviral particles were concentrated by polyethylene glycol 6000 precipitation as decribed before (*Kutner et al., 2009*). We dissolved the purified retroviral particles in PBS and stored working aliquots at 80°C. To determine the titres of viral preparations, NIH3T3 cells were transduced with tenfold serial dilutions of the purified retroviral particles by spinfection. We calculated functional titres based on the percentage of GFP-positive cells measured by flow cytometry as described before (*Kutner et al., 2009*).

## Surface staining for flow cytometry

Activated cells were incubated in anti-CD3ε and anti-CD28 at 37 °C for 20 min with 5% $CO_2$ while resting cells were kept on ice. The unbound antibody was then washed off with ice cold complete RPMI medium. Samples were washed with ice cold 0.5 M glycine pH 2.2 to remove any surface bound antibody and excessively washed with ice cold complete RPMI medium. All cells where then incubated with anti-CD28 FITC for 1 hr at 4 °C. Afterwards cells were washed with ice cold complete

RPMI medium and resuspended in Flow-Buffer. Samples were acquired using an LSR-II flow cytometer (Becton-Dickinson, San Jose, CA) and analysed using FlowJo software 10 (Tree Star, Ashland, OR) and GraphPad.Prism.

## Internalisation assay in flow cytometry

A total of $3 \times 10^5$ Jurkat T cells were incubated with anti-CD3ε and anti-CD28 FITC on ice for 1 hr. Unbound antibody was washed off with ice cold complete RPMI medium. Cells for the activated condition where incubated at 37 °C for 20 min with 5% $CO_2$ atmosphere, while the resting condition was kept on ice to make sure no endocytosis was occurring. All cells were then washed twice with acid to remove any surface bound antibody. Following, cells were washed twice with ice cold complete RPMI media and resuspended in FLOW-Buffer (2% FCS, 2 mM Edta, 0.02% sodium azide, 1xPBS). Samples were acquired using an LSR-II flow cytometer (Becton-Dickinson, San Jose, CA) and analysed using FlowJo software 10 (Tree Star, Ashland, OR) and GraphPad Prism.

## Intracellular staining in flow cytometry

Jurkat WT T cells and SNX9KO cells ($3 \times 10^5$) were activated in a 96 well plate coated with anti-CD3ε and anti-CD28 for 5 min. In resting conditions cells were kept on ice. Cells were then fixed with 3.7% paraformaldehyde for 20 min at either 37 °C or on ice (resting condition). Then the cells were washed 3 x with PBS and permeabilised with 0.15% triton-X100 for 10 min at RT. Cells were again washed 3 x with PBS and then stained with antibodies against phosphorylated CD28-pY218 (pCD28) or phosphorylated TCR ζ -pY142 (pTCR).

All flow-cytometry-based assays were performed on a BD FACSCanto II or a BD LSR II. The size of the population in FSC and SSC dot plot were used to distinguish live and dead cells. Doublets were eliminated by using a pulse geometry gate with FSC-H and FSC-A.

## Microscopy

Fixed, live-cell confocal and two-photon FRAP microscopy were performed on a Zeiss LSM880 laser-scanning confocal microscope (Zeiss, Germany; ZEN Digital Imaging for Light Microscopy, RRID:SCR_013672) which is equipped with an argon laser (405, 488 nm), a diode pump solid state laser (561, 647 nm), with a Mai Tai Insight DeepSee tuneable multi-photon laser and a live-cell incubation chamber (Pecon). Images were acquired with a 100 × 1.4 NA DIC M27 Apo-Plan oil immersion objective (Zeiss, Germany) and GaAsP-PMTs in simultaneous, bidirectional scanning mode, resulting in two-colour frame recording almost every 10 s. For each channel, the pinhole was set to 1 Airy Unit.

Fixed and live-cell TIRF images were acquired on a total internal reflection fluorescence microscope (ELYRA; Zeiss) with a 100× oil-immersion objective with a numerical aperture of 1.46. Recorded images were analysed with Zeiss ZEN Black software. GFP constructs were excited using the 488 nm line of the argon laser source, while PamCherry and mCherry tagged proteins were excited with the 561 nm laser line.

## Correlative light and electron microscopy

Cells co-transfected with SNX9-mCherry and CD28-EGFP were activated for 20 min on 35 mm in-plane gridded dishes (MatTek Corporation) coated with anti-CD3ε and anti-CD28 antibodies, fixed in 3.7% PFA. Bright field images as well as confocal z-stacks were obtained on a LSM880 laser-scanning confocal microscope using an Airyscan detector. The position of cells of interest were noted with the alphanumeric coordinate observable in brightfield imaging. Cells were then processed for transmission electron microscopy (TEM) by re-fixation with 2.5% glutaraldehyde in 0.1 M sodium cacodylate buffer (pH7.4) for 1 hr at room temperature. Cells were washed with 0.1 M sodium cacodylate buffer, post-fixed in 2% reduced osmium tetroxide ($OsO_4$), 1% thiocarbohydrazide and 1% $OsO_4$ in cacodylate buffer. Cells were en bloc stained in 2% uranyl acetate and lead aspartate and serially dehydrated in increasing percentages of ethanol at 30%, 50%, 70%, 90%, and 100%. Cells were then serially infiltrated with Durcupan Resin (Sigma 44610) at 33%, 66%, and 100% in a BioWave microwave (Pelco). Fresh 100% resin was then added and polymerised to hardness at 60°C for 48 hr. Thick sections (250 nm) were cut on an ultramicrotome (UC7: Leica Microsystems) and the first section was collected onto triple slot carbon grids (Pelco) to ensure the immunological synapse was within the section volume. Grids were coated with 10 nm fiducial gold particles and cells of interest were re-registered

on a 200 kV Thermo Fisher Talos Arctica operated at room temperature. High-magnification tiled micrographs were acquired using MAPS software (Thermo Fisher) on the whole cell of interest to find the area of interest positive for CD28-EGFP and SNX9-mCherry. Dual axis tilt series were collected at 1 degree increments from –60 to +60 degrees with a Falcon three camera operated in linear mode at binning of 1 using Tomography software (Thermo Fisher). Tomograms were reconstructed using weighted back-projection in IMOD.

Cells of interest were correlated using the alphanumeric code of MatTek in plane-gridded dishes (*Ariotti et al., 2018*). The first thick ultramicrotome sections were correlated to an individual confocal z-slice of the cell of interest based on the CD28 signal at the plasma membrane and re-registered using plasma membrane features as landmarks by electron microscopy. Overlays between CD28, SNX9 and montaged ×10,000 magnification whole cell transmission electron micrographs were generated, and features were aligned at high magnification using Puppet Warp in Photoshop (Adobe).

## Image analysis

SNX9 translocation to the immunological synapse was quantified through a blind analysis. Random fields of view of the stained (phalloidin and anti-SNX9) conjugates -SEE,+ SEE for 2, 5, 10, and 15 min were imaged, assigned an identification number and all stored in one folder on a hard drive. Immunological synapses were then identified by independent visual observation by three researchers based on the accumulation, on the phalloidin channel, of F-actin at the cell-cell interface, without opening the SNX9 channel of these images. The conjugates identified as forming a synapse based on this criterion were then marked (using the multi point tool form Image J). In a second step, the marked conjugates were evaluated for SNX9 accumulation by independent visual observation by the three researchers.

Jurkat-Raji B cell conjugate 3D reconstructions were made in Imaris (Imaris, RRID:SCR_007370). 3D reconstructions were then clipped manually in the orthogonal view in order to obtain the best possible en face projection of the interface between the two cell types. Once clipped the en face views were exported by making a snapshot.

Number of intracellular structures per stack was determined using FIJI (Fiji, RRID:SCR_002285) as described in *Figure 1—figure supplement 2* to *Figure 1*. 1. A threshold was manually applied to the original data to identify endosomal structures in the z-plane with maximal identifiable structures. The threshold was applied to the entire Z-stack to generate a subsequent binary endosome mask for each image in the stack. Finally, the endosome mask was analysed to identify endosomal structures above a minimum of 5-pixel units in total size (pixel size: 65 nm).

To analyse the overlap between clusters of two CD28 and SNX9, an intensity-based threshold was generated to identify endosomes/structures in each channel. This threshold was used to generate a mask for each channel, and the CD28 mask subtracted from the inverted SNX9 mask to generate a mask of the overlap between channels. The number of endosomes/structures present in the overlap mask was quantified and divided by the total number of CD28 endosomes/structures identified to determine the percentage of total CD28 endosomes overlapping with SNX9. A similar approach was used to measure incorporation of PAmCherry signal into GFP positive structures, whereby one threshold was used to define GFP-positive structures and one the cytosol, masks of each generated, and incorporation of PAmCherry signal in the cluster and cytosol threshold quantified. FIJI was also used to create histograms of fold fluorescent intensities.

To quantify the formation of endocytic vesicles after photoactivation of SNX9- or TCR$\zeta$-PAm-Cherry, images of the time series were smoothed, and an intensity-based threshold applied to identify endosomes and remove cytosolic signal, while ensuring no endosomes were identified pre-photoactivation. A size-based threshold was applied to single pixel or extremely large objects, and objects counted in each image frame.

The increase of fluorescence over time of the two-photon FRAP experiments was corrected for the bleaching resulting from the imaging laser during acquisition by normalisation with the signal from unbleached ROIs. The final recovery curve was obtained by calculating the mean of all the individual acquisitions. the time course of fluorescence recovery to these regions was measured using the ROI measurement tool from ZEN (ZEN Digital Imaging for Light Microscopy, RRID:SCR_013672).

For the fitting, fluorescence recovery after photobleaching (FRAP) curves were normalised by first subtracting the fluorescence intensity right after photobleaching, and by then dividing by the

pre-bleach steady-state fluorescence intensity, limiting the range of intensity values to between 0 and 1. Normalised curves were then fitted with either a mono-, bi-, or triple-exponential decay model, with the goodness-of-fit evaluated by the sum of the weighted summed square of residuals (SSR). The best result was obtained using a bi-exponential decay, reflecting that the fluorescence recovery occurred via two processes characterised by kinetic constants tau1 and tau2.

To analyse the amount of NFAT fluorescence over the nucleus in respect to NFAT fluorescence of the the whole cell (as a measure of nuclear translocation of NFAT), the ImageJ/Fiji Plugin Intensity_Ratio_Nuclei_Cytoplasm was used which is detailed at: http://dev.mri.cnrs.fr/projects/imagej-macros/wiki/Intensity_Ratio_Nuclei_Cytoplasm_Tool.

Electron tomograph images were reconstructed using IMOD 4.11 (https://bio3d.colorado.edu/imod/, RRID:SCR_003297). Rendered surfaces were generated in IMOD for tubules connected to the plasma membrane, and the length, surface area and volume of the surfaces quantified using the IMODINFO plugin. Tubules were defined based on 1- co-localisation with SNX9 and CD28 fluorescence signal; 2- originating from the plasma membrane and 3- electron density consistent with either other SNX9 tubules, endoplasmic reticulum tubules or clathrin-coated pits Specific tubules were excluded from being analysed as SNX9-positive tubules based upon their distinct densities.

## Statistical analysis

All statistical analysis excepting time of divergence analysis were performed using GraphPad software (Graphpad Prism, RRID:SCR_002798). Statistical significance between datasets was determined by performing two-tailed, unpaired non-parametric Students T-tests. Graphs show mean values, and error bars represent the SEM. In statistical analysis, $p > 0.05$ is indicated as not significant (n.s.), whereas statistically significant values are indicated by asterisks as follows: *$p \leq 0.05$, **$p < 0.01$, ***$p < 0.001$.

Time of divergence analysis was performed using a custom MatLab script as previously described (*Redpath et al., 2019*).

## Acknowledgements

General: We thank the staff of the BioMedical Imaging Facility of the University of New South Wales and the facilities supported by AMMRF at the Electron Microscope Unit at UNSW. Funding: We thank the funding bodies: National Health, Medical Research Council (APP1102730) the the Deutsche Forschungsgemeinschaft (RO 6238/1–1), the Swiss National Science Foundation (31,003 A_172969), the Thurgauische Stiftung für Wissenschaft und Forschung, the State Secretariat for Education, Research and Innovation and the Novartis Foundation for Biomedical Research.

## Additional information

### Funding

| Funder | Grant reference number | Author |
| --- | --- | --- |
| Schweizerischer Nationalfonds zur Förderung der Wissenschaftlichen Forschung | 31003A_172969 | Jeremie Rossy |
| Deutsche Forschungsgemeinschaft | RO 6238/1-1 | Jeremie Rossy |
| National Health and Medical Research Council | APP1102730 | Jeremie Rossy |
| Novartis Stiftung für Medizinisch-Biologische Forschung | | Jeremie Rossy |

The funders had no role in study design, data collection and interpretation, or the decision to submit the work for publication.

## Author contributions
Manuela Ecker, Data curation, Formal analysis, Investigation, Writing – original draft; Richard Schregle, Pascal Rossatti, Data curation, Formal analysis, Investigation; Natasha Kapoor-Kaushik, Data curation, Investigation; Verena M Betzler, Investigation, Supervision; Daryan Kempe, Formal analysis, Methodology, Visualization; Maté Biro, Supervision; Nicholas Ariotti, Formal analysis, Investigation, Methodology, Visualization; Gregory MI Redpath, Conceptualization, Data curation, Formal analysis, Investigation, Methodology, Supervision; Jeremie Rossy, Conceptualization, Formal analysis, Funding acquisition, Project administration, Supervision, Writing – original draft, Writing - review and editing

## Author ORCIDs
Maté Biro http://orcid.org/0000-0001-5852-3726
Jeremie Rossy http://orcid.org/0000-0002-5128-5283

## Decision letter and Author response
Decision letter https://doi.org/10.7554/eLife.67550.sa1
Author response https://doi.org/10.7554/eLife.67550.sa2

## Additional files

### Supplementary files
• Transparent reporting form

### Data availability
All datasets for this study are deposited on Zenodo and are publicly available under a Creative Commons Attribution 4.0 International license.

The following dataset was generated:

| Author(s) | Year | Dataset title | Dataset URL | Database and Identifier |
|---|---|---|---|---|
| Ecker M, Rossy J, Redpath GMI, Ariotti N | 2021 | SNX9-induced membrane tubulation regulates CD28 cluster stability and signalling | https://doi.org/10.5281/zenodo.5838426 | Zenodo, 10.5281/zenodo.5838426 |

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
