## [Editor Report]

Efficient T cell activation requires co-engagement of the T cell receptor (TCR) and of the CD28 co-stimulatory molecules. Both the TCR and CD28 form micro clusters at the plasma membrane of activated T cells. The present study is important in that it showed that by inducing membrane tubulation, the BAR domain-containing protein SNX9 generates a membrane environment that promotes CD28 triggering.

---

## [Decision Letter]

**Decision letter after peer review:**

Thank you for submitting your article "SNX9-induced membrane tubulation regulates CD28 cluster stability and signalling" for consideration by *eLife*. Your article has been reviewed by 3 peer reviewers, and the evaluation has been overseen by a Reviewing Editor and Tadatsugu Taniguchi as the Senior Editor. The following individuals involved in review of your submission have agreed to reveal their identity: Christoph Wülfing (Reviewer #1); Chenqi Xu (Reviewer #2).

Essential Revisions:

Four essential revisions are needed:

1. Corroboration in primary T cells

2. Corroboration with APCs

3. Provide some biochemical data on the CD28 SNX9 interaction

4. Improved data quantification

*Reviewer #1:*

The mutual interdependence of receptor localization and vesicular trafficking is of great interest, i.e. the linked questions of how receptors control vesicular trafficking that controls receptor localization. Here Ecker et al., link the T cell costimulatory receptor CD28 to the sorting nexin SNX9. They show that SNX9 is recruited to the vicinity of CD28 clusters formed when Jurkat T cells are activated on cover slips coated with anti-CD3 and anti-CD28 antibodies. This recruitment was dynamic and required the SNX9 SH3 and PX domains. Correlative light electron microscopy suggests that tubular structures are associated with CD28 and SNX9, something to be expected for the sorting nexin family. Overexpression of SNX9 led to more stable CD28 clusters on the antibody-coated cover slips. Deletion of SNX9 led to reduced CD28 phosphorylation, NFAT nuclear translocation and IL-2 secretion upon activation of Jurkat T cells by antibody-coated cover slips and Raji B cells, respectively. The strength of the manuscript lies in extensive, technically demanding and interesting imaging data on CD28 and SNX9 with the potential to establish intriguing biological connections. There are substantial conceptual and technical concerns. Briefly, corroboration of key data obtained using T cells activated with antibody-coated glass cover slips with T cells activated by antigen presenting cells are largely missing. Such corroboration is critical as fixing receptor location by antibody binding and limited membrane deformation on the cover slips substantially alter receptor localization, versicular trafficking and the connection between the two processes. In addition, incomplete image quantification limits the conclusions that can be drawn for the extensive imaging data.

1. Linking receptor localization and vesicular trafficking is technically highly demanding, as many of the relevant structures have a size below the resolution limit of fluorescence microscopy. The formation of receptor clusters is greatly enhanced when T cells are activated on antibody-coated cover slips and such clusters are much more readily detectable as immobile diffraction-limited objects in a single imaging plane. Using antibody-coated cover slips thus is an important tool in investigating receptor localization and vesicular trafficking at the T cell synapse. However, with receptor clustering and vesicular trafficking it is critical to confirm all key findings in cell couples between T cells and antigen presenting cells. Lateral mobility influences receptor localization (and is prevented on the cover slips by the immobile antibodies fixing the receptors in place) and membrane deformation is central to vesicular trafficking, in particular its regulation by BAR domain-containing proteins (as greatly reduced on the flat cover slips). Work from the Hivroz laboratory illustrates how cover slips and cell couples can be effectively used to complement each other. With the exception of Figure 1A (which doesn't resolve SNX9 puncta) and Figure 6G (IL-2 secretion) the data in the main figures rely on antibody-coated cover slips only. At the very least, the existence of co-localized CD28 and SNX9 puncta needs to be shown in cell couples.

2. Using images to understand biology is rewarding because of their high information content, yet it requires rigorous quantification. In particular any new structure of interest should be defined in quantitative terms rather than presented through representative images. This applies prominently to the CD28 'ridges' in Figure 2 and the CD28/SNX9-associated 'tubules' in Figure 4.

The order of recruitment of CD28 and SNX9 to the 'ridges' is an important part of the author's argument that SNX9 doesn't control CD28 clustering but the signaling of the clusters. For that case to be made convincingly, the authors should define quantitative criteria what constitutes a 'ridge', some idea of what the cell biological structure underlying them is would be good (first membrane contacts stabilized by binding of CD28 to the anti-CD28 antibody? IRM to show glass membrane contacts would be very helpful) and then recruitment of SNX9 and a negative control protein (GFP?) to the independently defined ridges would need to be quantified as a function of time.

Similarly, Figure 4 shows that tubules occur in regions where there is CD28 and SNX9. By visual inspection of the single representative image set, tubules also occur outside the areas of CD28 and SNX9 accumulation. The authors should identify tubules independently in the EM images (e.g. length, frequency) and then calculate whether they are enriched in areas of CD28 fluorescence, SNX9 fluorescence and areas positive for both. In addition, a detailed description of how the fluorescence and EM data were aligned would be welcome.

Additional figures would benefit from enhanced image quantification and/or a more detailed description of the methods to do so.

Criteria for interface enrichment of SNX9 in Figure 1A/B should be given.

Vicinity to the glass cover slip should be quantified in Figure 1C.

The methods relating to Figure 1E state that 'endosomes were first identified by manual thresholding'. The authors should describe why that was done, how that was done in detail and preferentially show supplementary data how this affects the figure. This is important as Figure 1E established a dominant role of CD28 in SNX9 recruitment.

Statistics for the differences in distances from the cover slip in Figure 1G would be welcome.

The colocalization between CD28 and SNX9 in Figure 2A/B should be quantified. This is important as the figure makes the argument that CD28 signaling is not required for SNX9 recruitment. However, the visual inspection of the two representative traces suggests better colocalization of CD28 with SNX9 than CD28YF. Inclusion of a negative control to establish a baseline of random colocalization would be welcome.

There are a number of additional comments:

The authors state that the SNX9-mCherry signal in Figure 1C was organized in 'membranous structures'. This reviewer can see punctuate structures. How do the authors know that the structures are membranous, rather than e.g. large protein complexes?

Poly L-lysine is activating for T cells (Nature Immunology 19, 203-205 (2018)). This should be acknowledged.

With regard to Figure 3A the authors state that they 'did not observe formation of any endosomal structures positive for SNX9-PAmcherry'. If they would like to make the statement, it should be supported by quantitative data.

In Figure 5E/F the authors have chosen to quantify fluorescence recovery at 380s showing a slight delay upon SNX9 knockout. At earlier time points up to 240 s they would reach a partially different conclusion, as one of the two knock out cell lines behaves the same as the control. This should be acknowledged, and conclusions should be modified accordingly. In the same experiment it would be good to know how much overexpression of SNX9 changes its expression level relative to endogenous protein expression.

As overexpression of SNX9 rather than its knockout has a substantial effect on CD28 complex stability, it would be good to see what the function effects thereof (NFAT, IL-2) are.

*Reviewer #2:*

In the manuscript, Ecker et al., designed serial experiments to address the function of a BAR-domain protein SNX9 in regulating CD28 signaling. In addition to the BAR domain that senses membrane curvature, SNX9 also contains a PX domain to interact with phosphoinositides and a SH3 domain to interact with membrane-remodeling proteins. The physiological functions of SNX9 in T cells are not fully understood.

The major strength of this manuscript is its application of multiple imaging technologies in measuring CD28-SNX9 interaction, which can be further strengthened by additional biochemical experiments. Through the knockout strategy, the authors showed that SNX9 promotes CD28 signaling, which agrees with its role in promoting CD28 clustering. The physiological importance of SNX9 can be further established by applying other manipulation strategies including SNX9 overexpression and mutations.

This manuscript highlights a fact that different immunoreceptors are subject to specific regulations. Understanding the detailed mechanisms can pave the way for developing new immunotherapies based on precise manipulation of immunoreceptor signaling.

Major comments:

1. CD28-SNX9 interaction needs to be further confirmed. In addition to the nice imaging experiments showing co-localization between CD28 and SNX9, biochemical experiments such as co-immunoprecipitation will be helpful to confirm the interaction and the key interaction sites. Or if the interaction is rather indirect, can the authors provide the bridging mechanism?

2. The regulation of CD28 clustering by SNX9 needs to be further studied. The authors suggest that SNX9 contributes to regulate the rates of exchange of CD28 in and out of clusters and thereby their stability. If that is the case, CD28 clustering dynamics, in particularly at later time points when clusters disassemble, should be affected by manipulation of SNX9 levels. In addition to the snapshot images shown in Figure 5A, real-time TIRF imaging should be more informative to show the impacts of SNX9 manipulation. The authors proposed that SNX9 might regulate local lipid environment to stabilize CD28 clustering. This idea can be tested in a straightforward way by using widely used lipid probes. Moreover, the binding between CD28 cytoplasmic domain with the plasma membrane can also be imaged when SNX9 levels are altered.

3. The function of SNX9 in CD28 signaling needed to be further demonstrated. In addition to the SNX9 knockout strategy, the SNX9 overexpression strategy and mutation strategy (mutating PX or SH3 domain) should be applied to see the functional outcomes of CD28 signaling.

4. In terms of CD28 internalization, can the authors comment why they observe opposing results as Badour et al., (PNAS 2007)? In Figure S5, it seems that SNX9-KO#4 cells had higher levels of CD28 internalization.

*Reviewer #3:*

The paper by Ecker et al., elegantly documents the role of SNX9 in CD28 clustering. The analysis is at a high level and the paper should be of interest to investigators in the field. The major shortcoming is the focus on the use of Jurkat T-cells as a model for their work. This choice is a puzzle since Jurkat cells have been well documented to have major defects in key signaling pathways linked to CD28 function such as a lack of PTEN which has major effect on the expression level of D-3 lipids which play in role in SNX9 function. The paper needs to include more data on primary T-cells and if possible from confirmation of their findings in T-cells interacting with normal antigen presenting cells and in response to the natural ligands CD80/CD86. There are many TCR transgenic mouse models which respond to specific antigens presented by dendritic cells. This is a normal expectation and would greatly strengthen the paper. The authors should also address there any evidence of binding in SNX9-CD28 binding in their studies (i.e. co-IP?) and how the data fits with the AP2 complex and PKCtheta interations with CD28. There are also some issues related to sample size and statistics that need to be addressed in one figure.

Specific Points

1) The paper needs to include more data on primary T-cells and if possible from confirmation of their findings in T-cells interacting with normal antigen presenting cells. There are many TCR transgenic mouse models which respond to specific antigens presented by dendritic cells. This is a normal expectation and would greatly strengthen the paper.

2) The implications related to CD28 endocytosis seem important. In Figure S5, the authors present some data but it is puzzling. Firstly, the behaviour of KO#3 vs KO#4 seem different…why? If there are differences between the two Kos, is this not a problem for the other analysis? Secondly, there seem only 2 points for WT making the statistical analysis unclear. Some mention of the observation that YMXM mutations affect CD28 π 3K binding endocytosis should be made (PMID: 9498761).

3) Is there any evidence of binding in SNX9-CD28 binding in their studies (i.e. co-IP?) and is the AP2 complex part of the cluster or does this binding occur after the involvement of SNX9?

4) It is unclear whether the connection between SNX9 and CD28 is unique or is CD28 simply a surrogate for other receptors such as the TCR complex? It is too bad that CTLA-4 is not present…this is be very exciting if CTLA-4 disrupted the link between SNX9 and CD28 as might be seen with pre-activated T-cells?

5) Figure 1: analysis is done on slides with anti-CD3 and CD28; however it is important to show some analysis with immobilised CD80 or CD86 to be certain that the effects seen are not due to some artificial effect of high affinity antibody rather than natural ligand. It is not necessary to do all the analysis with natural ligand but at least to replicate some of the key features.

6) It is still not clear show the clustering specifically relates to the formation of the c-SMAC and p-SMAC…do the authors have images that include LFA-1 as a marker for the pSMAC? PKCtheta localisation is connected to CD28 but is SNX9 connected to PKCtheta which in turn leads to CD28…what is the model?

7) Figure 6: Panel C..which pTCR? Zeta and CD3 ITAMs? Panel E: the authors need to explain more clearly what their standard is for NFAT nuclear localisation (i.e. 50% is scored positive?) It is also noteworthy that some cells in the SNX9KO overlap with wild type cells…is there an explanation? Is it that T-cells are heterogenous in their dependency on SNX9-CD28…this could be quite interesting if true?

8) Figure S4: small point…no impact on Lck, but do tyrosine kinase inhibitors interfere with SNX9-CD28..is the process tyrosine phosphorylation dependent? An easy experiment to do.

---

## [Author Response]

Essential revisions:Four essential revisions are needed:1. Corroboration in primary T cellsReviewer 3:– The paper needs to include more data on primary T-cells and if possible from confirmation of their findings in T-cells interacting with normal antigen presenting cells. There are many TCR transgenic mouse models which respond to specific antigens presented by dendritic cells. This is a normal expectation and would greatly strengthen the paper.

We agree with the reviewer. Even if Jurkat T cells are an extensively validated model of T cell activation and immunological synapse formation, primary cell data was missing from the original submission. As suggested, we have used a transgenic mouse model to validate the co-recruitment to the immunological synapse of CD28 and SNX9. We used primary CD4 T cells isolated from OT-II transgenic mice, which were activated with LPS-matured mouse bone marrow derived dendritic cells (BMDCs) pulsed with the OT-II peptide. BMDCs express high levels of SNX9, which prevented clear visualisation of SNX9 recruitment to the T cell synapse by immunofluorescence. We therefore virally transduced the OT-II T cells so that they expressed SNX9-GFP and detected CD28 with an antibody.

These experiments confirmed the data obtained in Jurkat cells (new Figure 1—figure supplement 1). SNX9 was unambiguously recruited to the immunological synapse of CD4 T cell, where it co-localised with CD28. These images further show that SNX9 was cytosolic in primary CD4 T cells interacting with BMDCs that did not present the OT-II peptide, confirming that SNX9 is recruited to membranous structures only in activated T cells. The main text has been amended to include these results (lines 122-130, 140-142, 218-220).

2. Corroboration with APCsReviewer 1:– The formation of receptor clusters is greatly enhanced when T cells are activated on antibody-coated cover slips and such clusters are much more readily detectable as immobile diffraction-limited objects in a single imaging plane. Using antibody-coated cover slips thus is an important tool in investigating receptor localization and vesicular trafficking at the T cell synapse. However, with receptor clustering and vesicular trafficking it is critical to confirm all key findings in cell couples between T cells and antigen presenting cells. Lateral mobility influences receptor localization (and is prevented on the cover slips by the immobile antibodies fixing the receptors in place) and membrane deformation is central to vesicular trafficking, in particular its regulation by BAR domain-containing proteins (as greatly reduced on the flat cover slips). Work from the Hivroz laboratory illustrates how cover slips and cell couples can be effectively used to complement each other. With the exception of Figure 1A (which doesn’t resolve SNX9 puncta) and Figure 6G (IL-2 secretion) the data in the main figures rely on antibody-coated cover slips only. At the very least, the existence of co-localized CD28 and SNX9 puncta needs to be shown in cell couples.

We fully understand the reviewer’s concern regarding the immobilisation of activating antibodies on cover glass. As suggested, we performed experiments in conjugates, between Jurkat T cells and Raji cells, to confirm the co-clustering of SNX9 and CD28 in a system allowing ligand mobility and membrane deformation.

By acquiring high-resolution z-stacks using Airy scan microscopy, we were able to confirm the existence of puncta where CD28 and SNX9 clearly co-localised. These are visible in the enface views of 3D reconstructions of Z-stacks. Importantly, such double-positive puncta were not visible in conjugate where Raji B cells had not been pulsed with super antigen. These data are now included in Figure 2—figure supplement 3 and mentioned in the main text (lines 213220).

– Poly L-lysine is activating for T cells (Nature Immunology 19, 203-205 (2018)). This should be acknowledged.

We agree with the reviewer and now cite the mentioned study in the manuscript (lines 142143). We would like to point out however that our data indicate no obvious activating properties of poly-L-lysine (PLL). Typically, SNX9 was fully cytosolic in cells on PLL while completely recruited to membrane structures in cells activated by antibodies or antigenpresenting cells (Figure 1D). Similarly, we observed no nuclear translocation of NFAT in cells plated on PLL (Figure 7E).

Reviewer 3:– Figure 1: analysis is done on slides with anti-CD3 and CD28; however it is important to show some analysis with immobilised CD80 or CD86 to be certain that the effects seen are not due to some artificial effect of high affinity antibody rather than natural ligand. It is not necessary to do all the analysis with natural ligand but at least to replicate some of the key features.

We understand the concern of the reviewer and apologise for the confusion as we did not intend to suggest that clustering of CD28 is part of the novel findings presented in this manuscript. Indeed, clustering of CD28 at the immunological synapse has been extensively demonstrated in the past (typically in the work of the groups of T. Saito and M. Dustin, which we cite lines 53, 201 and 527).

Nevertheless, we have indirectly addressed this point in the new data shown in Figure 2—figure supplement 3, which shows puncta at the immunological synapse that are positive for CD28 and SNX9, strongly suggesting that CD28 and SNX9 co-cluster in response to natural ligands. We also would like to point out that one of the functional consequences of SNX9 knock-out – secretion of IL-2 – has been investigated using stimulation by antigen-presenting cells, which further indicates that the importance of SNX9 for CD28 signalling is not only observable in the context of CD28 cross-linking by high-affinity antibodies onto an immobilised surface.

3. Provide some biochemical data on the CD28 SNX9 interactionReviewer 2:– CD28-SNX9 interaction needs to be further confirmed. In addition to the nice imaging experiments showing co-localization between CD28 and SNX9, biochemical experiments such as co-immunoprecipitation will be helpful to confirm the interaction and the key interaction sites?Reviewer 3:– Is there any evidence of binding in SNX9-CD28 binding in their studies (i.e. co-IP?)

To investigate a possible direct interaction between CD28 and SN9, we performed immunoprecipitation experiments as suggested by the reviewers (Author response image 1). However, we were not able to detect CD28 when we immunoprecipitated SNX9-GFP expressed in SNX9KO cells, using either an anti-SNX9 or a GFP trap. Similarly, we detected no SNX9 after immunoprecipitation with an antibody against CD28. These data support the hypothesis presented in the discussion, which postulates that the SNX9 contribution to CD28 signalling is probably to generate a favourable membrane environment (curvature or charge), as opposed to relying on a direct physical interaction between the two proteins.

**Author response image 1. sa2fig1:** Co-immunoprecipitation (Co-IP) analysis of the interaction between SNX9 and CD28. SNX9knock-out Jurkat T cells were retrovirally reconstituted to stably express GFP-SNX9. The Jurkat T cells were activated or not for 20-30 min with anti-CD3ε (OKT3) and anti-CD28 (CD28.2) antibodies bound onto dished additionally coated with PLL (Σ Aldrich) and human fibronectin (in-house). After lysis, cell lysates were subjected to Co-IP and WB analysis. (A) CD28 and GFP-SNX9 were immunoprecipitated using anti-CD28 (D2Z4E; Cell Signaling Technology), anti-SNX9 (EPR14399; Abcam) and GFPtrap (Chromotek), blotted onto nitrocellulose, and detected using the same antibodies (CD28 and SNX9) and anti-GFP (E385; Abcam) antibodies. When GFP-SNX9 was immunoprecipitated with either SNX9 or GFP antibodies, no CD28 coimmunoprecipitated (bottom blot, left and right panel). The same was observed when CD28 was immunoprecipitated, where no GFP-SNX9 co-immunoprecipitated (top and middle blot, middle panels). We did not observe an effect on the immunoprecipitation of CD28 and GFP-SNX9 by activation of the Jurkat T cells. The star (*) denotes heavy chains from the antibodies used for the Co-IP. (B) Load samples were analysed by WB to ensure proper expression of the target proteins. GFP-SNX9 was detected only in reconstituted cells (top blots). CD28 was detected in all samples (middle blot). Equal loading confirmed by anti-βActin antibodies (AC-15; Abcam). One representative experiment of three experiments with the same results is shown.

4. Improved data quantificationReviewer 1:– Criteria for interface enrichment of SNX9 in Figure 1A/B should be given.

We thank the reviewer for noticing that we had not provided enough information about how we evaluated SNX9 recruitment to the immunological synapse of these conjugates. The corresponding paragraph of the methods has been amended accordingly (lines 790-798).

– Vicinity to the glass cover slip should be quantified in Figure 1C.

We agree with the reviewer, we should not mention distances to the cover glass as we did not strictly obtain this measurement. We actually defined the 0 μm axial position (previously referred to as “cover glass”) as the first focal plane where the cells came in to focus. We have now amended the text to avoid confusion as to what we measured and be clearer about how we defined the reference point in these experiments (lines 172-182).

CD28 is at the plasma membrane and measuring the distance between the cell plasma membrane and a cover glass on which the cell stands is technically complex and beyond the resolution of Airyscan confocal microscopy. Typically, cell-substrate distances are in a range of 40 to 100 nm as measured with surface plasmon resonance microscopy

(https://doi.org/10.1016/j.optcom.2017.10.001 or https://doi.org/10.1016/S00063495(99)77219-X). Furthermore, we must disagree with the reviewer regarding the relevance of such a measurement in the context of the submitted manuscript. The point of Figure 1F and G is to show that SNX9 structures are above the plasma membrane (CD28 clusters) but under sub-synaptic endosomes defined by TCR or intracellular compartments defined by Rab5.

In summary, knowing the distance between CD28 molecules at the plasma membrane and the cover glass would not only be extremely challenging technically but it would also provide no significant insight into how SNX9 is spatially organised in activated T cells.

We have however performed a statistical test on these data as suggested in the next comment.

– Statistics for the differences in distances from the cover slip in Figure 1G would be welcome.

We agree that these statistics were missing and have extracted the axial position for the highest number of structures for each cell and evaluated the statistical significance of the difference between these positions. The resulting analysis is a new panel in Figure 1 (H). This analysis confirms that SNX9 intracellular structures are below TCR and Rab5-positive endosomes. However, while SNX9 maximum peaks above CD28 maximum, this difference is not statistically significant, and we have modified the main text accordingly (lines 182-187). Of note, the lack of statistical significance is most likely because of the signal of the SNX9positive tubules extends from 0 to 250 nm from the plasma membrane (new Figure 4—figure supplement 1, EM quantification), which prevents distinguishing between these tubules and CD28 clusters in the plasma membrane using Airy scan confocal microscopy.

– The methods relating to Figure 1E state that ‘endosomes were first identified by manual thresholding’. The authors should describe why that was done, how that was done in detail and preferentially show supplementary data how this affects the figure. This is important as Figure 1E established a dominant role of CD28 in SNX9 recruitment.

We thank the reviewer for noticing that we had not sufficiently described how an intensitybased threshold was used to identify the intracellular structures in Figure 1E. For better clarity, we have added a new figure supplement (2) to Figure 1 that describes step by step the workflow used in ImageJ to quantify these structures, which we mention in the main text line 138. We also have amended the methods to be more accurate as to how we have performed this quantification (lines 803-808).

– The colocalization between CD28 and SNX9 in Figure 2A/B should be quantified. This is important as the figure makes the argument that CD28 signaling is not required for SNX9 recruitment. However, the visual inspection of the two representative traces suggests better colocalization of CD28 with SNX9 than CD28YF. Inclusion of a negative control to establish a baseline of random colocalization would be welcome.

We agree with the reviewer, this quantification was missing and so was a negative control (no expected colocalization). We used a similar intensity-based threshold approach as now described in Figure 1—figure supplement 2 to generate a mask of the overlap between the GPF (CD28) and Alexa-647 (SNX9) channels. We have modified the methods to add a description of this analysis (lines 809-817) and the main text to mention its outcome (lines 203-209). These results are now shown in a new figure panel (Figure 2C) and new supplement (Figure 2—figure supplement 1). The reviewer’s visual inspection was confirmed by our analysis, as there is a clear tendency to a lower overlap between SNX9 clusters and the YF mutant than with WTCD28, although the difference was not significant. Nevertheless, it suggests that phosphorylation of CD28 contributes to recruit SNX9 to CD28 clusters, which we would not have been aware of without the reviewer’s suggestion.

– With regard to Figure 3A the authors state that they ‘did not observe formation of any endosomal structures positive for SNX9-Pamcherry’. If they would like to make the statement, it should be supported by quantitative data.

We thank the reviewer for noting the lack of relevant quantification. To overcome this oversight, we applied again an intensity-based threshold to identify the number of vesicles after photoactivation of TCRζPA-mCherry and SNX9-PamCherry (described in the methods, lines 819-823). This quantification, which confirms our statement about the absence of SNX9positive vesicles following photoactivation, is mentioned line 300-302 and is now a new supplement (Figure 3—figure supplement 2).

– Using images to understand biology is rewarding because of their high information content, yet it requires rigorous quantification. In particular any new structure of interest should be defined in quantitative terms rather than presented through representative images. This applies prominently to the CD28 ‘ridges’ in Figure 2 and the CD28/SNX9-associated ‘tubules’ in Figure 4.

We respectfully disagree with the reviewer concerning the ridges in Figure 2. These structures are commonly observed in T cell activated on antibody coated cover glassed and have been reported and described before, typically in the paper by Sherman et al., which we cite in the main text line 225. We actually mentioned these ridges only because they had been reported before in the context of T cells being dropped on antibody-coated glass, merely to indicate that we observed what others had observed before. If the reviewer considers that these structures should nevertheless be thoroughly quantified, we will happily remove any reference to them, as nowhere else in the manuscript we mentioned these ridges and they play no role in the model we put forward.

– Similarly, Figure 4 shows that tubules occur in regions where there is CD28 and SNX9. By visual inspection of the single representative image set, tubules also occur outside the areas of CD28 and SNX9 accumulation. The authors should identify tubules independently in the EM images (e.g. length, frequency) and then calculate whether they are enriched in areas of CD28 fluorescence, SNX9 fluorescence and areas positive for both. In addition, a detailed description of how the fluorescence and EM data were aligned would be welcome.

Following the reviewer’s suggestion, we have analysed the EM images using IMOD. We identified individual tubules based on (1) co-localisation with SNX9 and CD28 fluorescence signal; (2) connection to the plasma membrane and (3) electron density consistent with either other SNX9 tubules, endoplasmic reticulum tubules or clathrin-coated pits. We next quantified the length, surface area and volume of these tubules. The images showing the quantification approach and its results are now shown in Figure 4—figure supplement 1. These results indicate a clear tendency for tubules originating from CD28-rich area of the plasma membrane to be SNX9-positive and longer than other tubules identified at the immunological synapse. This has been added to the description of the EM data, lines 353-361 and to the methods lines 840-847.

We also have added a more detailed description of how the fluorescence and EM data were aligned in the methods (lines 783-788).

– In Figure 5E/F the authors have chosen to quantify fluorescence recovery at 380s showing a slight delay upon SNX9 knockout. At earlier time points up to 240 s they would reach a partially different conclusion, as one of the two knock out cell lines behaves the same as the control. This should be acknowledged, and conclusions should be modified accordingly. In the same experiment it would be good to know how much overexpression of SNX9 changes its expression level relative to endogenous protein expression

The reviewer is correct, the recovery curve of the KO#4 diverges from the WT only at later times points. We did acknowledge this in the original version of the manuscript (line 398).

We were however sensitive to the reviewer’s scepticism about the divergence at 380 sec and have performed further analyses to strengthen our conclusion. We have fitted the normalized recovery curves with bi-exponential decay model (with the goodness-of-fit confirmed by the sum of the weighted summed square of residuals). The kinetic constant characterizing the slow recovery process tau2 of both KO#3 and KO#4 differed significantly from the WT ones, confirming that fluorescence recovery is different in the WT than in both knock-out cell lines. Together with our original analysis, we believe there is better recovery in fluorescence in both KO cells and would prefer not to modify our conclusions.

We have added a reference to the results of this analysis in the main text (lines 399-405), a detailed description of how it was performed in the methods (lines 829-835) and show the charts corresponding to this analysis in a new Supplement (Figure 5—figure supplement 1).

Finally, we understand the reviewer’s concern regarding CD28 expression in the Kos, as it might indeed affect the recovery curves. However, data in the original version of the manuscript, Figure 6, indicated that CD28 expression (measured by flow cytometry) is not affected by the absence of SNX9. We have amended the main text to refer to this figure when we describe the FRAP data (lines 406-408).

Reviewer 3:– Figure 6: Panel C..which pTCR? Zeta and CD3 ITAMs?Panel E: the authors need to explain more clearly what their standard is for NFAT nuclear localisation (i.e. 50% is scored positive?).

The main text mentions the subunit of the TCR complex that is targeted by this antibody (TCRζ-Y142, line 483). However, the reviewer is correct, this was missing in the figure This has been corrected (lines 492).

We further apologise for the insufficient description of the way we quantified NFAT nuclear translocation. We did not use a scoring approach but only integrated NFAT fluorescence intensity over the Hoechst-stained nucleus. What is plotted in 7F is the percentage of the total NFAT fluorescence intensity measured in nuclei in relation to the total NFAT fluorescence detected. We have amended the figure legend (lines 496-498), the main text (line 511) and the methods (lines 836-837) to avoid that confusion.